# Giant isotropic magneto-thermal conductivity of metallic spin liquid candidate Pr$_2$Ir$_2$O$_7$ with quantum criticality

J. M. Ni[1], Y. Y. Huang[1], E. J. Cheng[1], Y. J. Yu[1], B. L. Pan[1], Q. Li[1], L. M. Xu[2], Z. M. Tian [2✉] & S. Y. Li [1,3,4✉]

Spin liquids are exotic states with no spontaneous symmetry breaking down to zero-temperature because of the highly entangled and fluctuating spins in frustrated systems. Exotic excitations like magnetic monopoles, visons, and photons may emerge from quantum spin ice states, a special kind of spin liquids in pyrochlore lattices. These materials usually are insulators, with an exception of the pyrochlore iridate Pr$_2$Ir$_2$O$_7$, which was proposed as a metallic spin liquid located at a zero-field quantum critical point. Here we report the ultralow-temperature thermal conductivity measurements on Pr$_2$Ir$_2$O$_7$. The Wiedemann–Franz law is verified at high fields and inferred at zero field, suggesting no breakdown of Landau quasi-particles at the quantum critical point, and the absence of mobile fermionic excitations. This result puts strong constraints on the description of the quantum criticality in Pr$_2$Ir$_2$O$_7$. Unexpectedly, although the specific heats are anisotropic with respect to magnetic field directions, the thermal conductivities display the giant but isotropic response. This indicates that quadrupolar interactions and quantum fluctuations are important, which will help determine the true ground state of this material.

[1] State Key Laboratory of Surface Physics, Department of Physics, and Laboratory of Advanced Materials, Fudan University, Shanghai 200438, China. [2] School of Physics, and Wuhan National High Magnetic Field Center, Huazhong University of Science and Technology, Wuhan 430074, China. [3] Collaborative Innovation Center of Advanced Microstructures, Nanjing 210093, China. [4] Shanghai Research Center for Quantum Sciences, Shanghai 201315, China. ✉email: tianzhaoming@hust.edu.cn; shiyan_li@fudan.edu.cn

Spin ice state on a frustrated pyrochlore lattice has attracted numerous interests in condensed matter physics, due to the emergent magnetic monopole excitations from the manifold of degenerate ground states[1,2]. By introducing quantum fluctuations with $J_{eff} = 1/2$ moments, quantum spin ice (QSI) states can be stabilized, exhibiting quantum electrodynamics with extra excitations like photons and visons[3]. (In this paper, we adopt the naming convention that the magnetic monopoles refer to the spin-flip excitation, while the visons refer to the sources of emergent electric fields[3].) $Yb_2Ti_2O_7$, $Tb_2Ti_2O_7$, and $Pr_2Zr_2O_7$ are such promising QSI candidates[3]. On the other hand, iridates with $5d$ electrons have also drawn much attention in recent years owing to the various quantum phases and transitions therein, which originate from the competition between spin-orbit coupling and electron–electron correlation[4,5]. When these two aspects meet in the pyrochlore iridate $Pr_2Ir_2O_7$, complex phenomena and exotic phases emerge[6–13].

$Pr_2Ir_2O_7$ is a metal with the antiferromagnetic RKKY interaction of about 20 K in non-Kramers Pr $4f$ moments mediated by Ir $5d$ conduction electrons[6]. The Kondo effect leads to a partial screening of the Pr $4f$ moments and gives a lower Weiss temperature of $|\theta_W| = 1.7$ K (ref. [6]). No long-range magnetic order was observed down to 70 mK evidenced by the magnetic susceptibility measurement, indicating a possible metallic spin liquid ground state, or even a U(1) QSI state[6,13]. A huge and anisotropic anomalous Hall effect (AHE) was probed under magnetic fields[7,9], which may be the result of the spin chirality effect on the Ir sites from the noncoplanar spin texture of Pr $4f$ moments. The observation of AHE in the absence of uniform magnetization at zero field further indicates a long-sought chiral spin liquid state in $Pr_2Ir_2O_7$ (ref. [8]). More interestingly, a zero-field quantum critical point (QCP) was uncovered from the diverging behavior and

scaling law in the Grüneisen ratio measurement[10]. It was also theoretically investigated as a QCP between antiferromagnetic ordering and nodal non-Fermi liquid[11]. Apparently, multiple mechanisms govern the ground state of $Pr_2Ir_2O_7$, which may induce rich phenomena beyond the spin-ice physics.

For such an exotic metallic spin-liquid candidate with quantum criticality, although various efforts have been made, two main issues remain to be solved. First, how do the electrons behave at the QCP? In other words, will the electrons still be well-defined Landau quasiparticles? Second, probably due to the large neutron absorption cross-section of the iridium ions and the very small size of its single crystals[14], little information is known for possible exotic magnetic excitations in $Pr_2Ir_2O_7$, from which knowledge about the role of multipolar interactions beyond the dipolar interactions and quantum fluctuations can be obtained.

Ultralow-temperature thermal conductivity measurement is an important technique to address the above two issues. For the former one, the verification of the Wiedemann–Franz (WF) law $\kappa/\sigma T = \pi^2 k_B^2/3e^2 = L_0$ can be viewed as an evidence of the survival of Landau quasiparticles at the QCP. Anomalous reduction of the Lorenz ratio $L(T)/L_0$ with $L(T) = \kappa/\sigma T$ has been observed in $CeCoIn_5$ (ref. [15]), $YbRh_2Si_2$ (under debate (refs. [16–18])), and YbAgGe (ref. [19]), while in some other compounds such as $CeNi_2Ge_2$ (ref. [20]) and $Sr_3Ru_2O_7$ (ref. [21]), the WF law is verified at the QCP. For the latter one, a sizable residual linear term of thermal conductivity indicates the presence of highly mobile gapless excitations in triangular organics $EtMe_3Sb[Pd(dmit)_2]_2$ (ref. [22]) (note that this result has been challenged by two recent reports[23,24]). Spinon thermal conductivity with a linear temperature dependence was also found in the ideal spin-1/2 antiferromagnetic Heisenberg chain copper benzoate[25]. No magnetic thermal conductivities were observed in other two QSL candidates $\kappa$-(BEDT-TTF)$_2$Cu$_2$(CN)$_3$ and YbMgGaO$_4$ (refs. [26,27]).

In this paper, we report ultralow-temperature thermal conductivity measurements on single crystals of $Pr_2Ir_2O_7$. The WF law is verified at high fields and inferred at zero field, suggesting the normal behavior of electrons at the QCP and the absence of fermionic magnetic excitations. A giant magneto-thermal conductivity at finite temperature is found, which may result from the strong scattering of phonons by the transverse fluctuations. The thermal conductivity is isotropic in different magnetic field directions, which is contrary to specific heat. We shall discuss the implications of these results.

## Results

**Charge and heat transport**. Figure 1a shows the temperature dependence of the resistivity $\rho(T)$ at zero field for the $Pr_2Ir_2O_7$ single crystal. The upturn behavior and the $\ln T$ dependence below 45 K where the resistivity displays a minimum, and the well fit to the Hamann's equation are the evidences for Kondo effect, as shown in the inset of Fig. 1a. This is consistent with ref. [6]. The magnetoresistance MR $= (\rho(H) - \rho(0\,T))/\rho(0\,T) \times 100\%$ at $T = 0.34$ K is presented in Fig. 1b. It is quite small, less than 5% up to 9 T, indicating the little influence of magnetic field on the charge transport. Note that no anisotropy of resistivity is reported with respect to the electric current direction[6], while the magnetoresistance is anisotropic in different magnetic field directions[9]. In the inset of Fig. 1b, $\rho(T)$ below 1 K in $\mu_0H = 0$, 3, and 6 T are plotted. Since all the curves are very flat, we can safely extrapolate them to the zero-temperature limit and get the residual resistivity $\rho_0 = 776$, 757, and 769 $\mu\Omega$ cm for $\mu_0H = 0$, 3, and 6 T, respectively.

The thermal conductivities of $Pr_2Ir_2O_7$ single crystal up to 7 T are shown in Fig. 2a. The magnetic fields were applied perpendicular to the (111) plane. At high fields like 5 T and 7

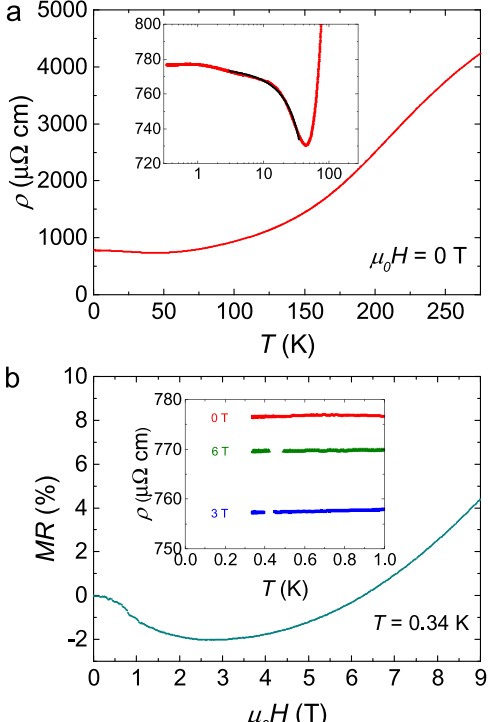

**Fig. 1 Charge transport results of Pr$_2$Ir$_2$O$_7$. a** Temperature dependence of the resistivity at zero field for Pr$_2$Ir$_2$O$_7$ single crystal. Inset: zoomed view of the resistivity minimum at 45 K due to the Kondo effect. The solid line is the fit to Hamann equation between 3 K and 35 K. **b** The magnetoresistance at $T = 0.34$ K. The magnetic field is applied along the [111] direction. Inset: $\rho(T)$ below 1 K in $\mu_0H = 0$, 3, and 6 T.

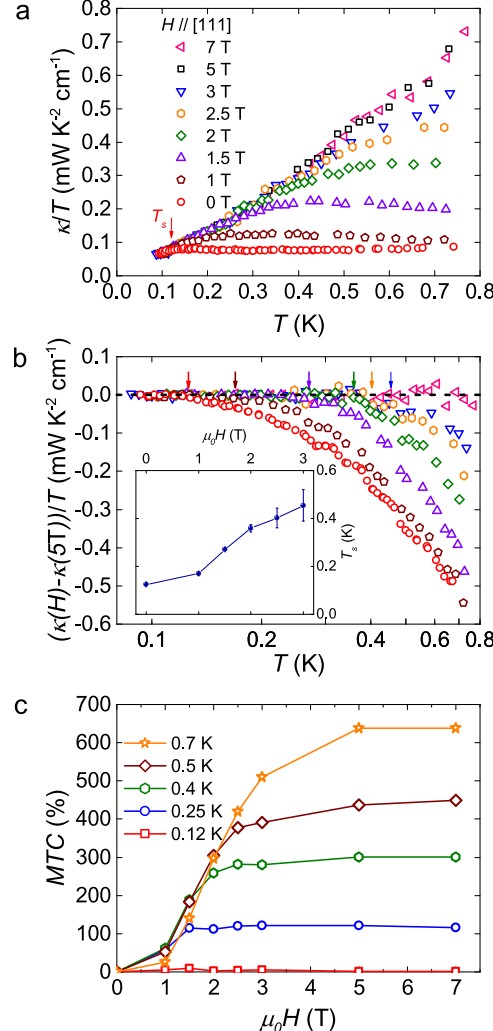

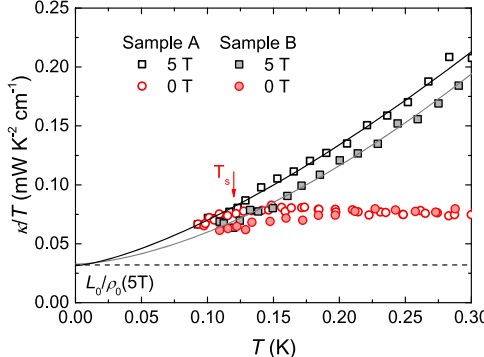

**Fig. 3 Verification of Wiedemann–Franz Law in $Pr_2Ir_2O_7$.** The thermal conductivity of two $Pr_2Ir_2O_7$ single crystals at $\mu_0H = 0$ and 5 T along the [111] direction, respectively. Solid lines are the fits of the thermal conductivity data to $\kappa/T = a + bT^{\alpha-1}$ at 5 T below 0.3 K. The dashed line is the Wiedemann-Franz law expectation $L_0/\rho_0(5\,T) = 0.032$ mW $K^{-2}$ $cm^{-1}$ for sample A at 5 T, which meets the extrapolated $\kappa/T \equiv a = 0.031$ mW $K^{-2}$ $cm^{-1}$ very well. The overlap of the 0 and 5 T curves below $T_s \approx 0.12$ K suggests that the Wiedemann-Franz law is also satisfied in zero field.

transport at 0.34 K, the MTC is as large as 100% at 0.25 K and even 650% at 0.7 K. For other QSL candidates such as $\kappa$-(BEDT-TTF)$_2$Cu$_2$(CN)$_3$ (ref. [26]) and YbMgGaO$_4$ (ref. [27]), there is also a positive MTC, but the magnitude is much smaller. Unexpectedly, a crossover from the positive MTCs at low temperatures to the negative MTCs at high temperatures is observed at $\theta_w$ energy scale (for details, see Supplementary Note 4). We will come back to discuss the origin of this giant MTC later.

We would like to emphasize that besides the electron thermal conductivities, $\kappa$ above 5 T is entirely due to phonons without magnetic scatterings. First, a metamagnetic transition at $B_c \sim 2.3$ T can be induced only when applying the field along the [111] direction in the magnetization $M(B)$ measurements[8]. This means that a sizeable fraction of the "2-in, 2-out" configurations are transformed into the "3-in, 1-out" configurations at the critical field. Therefore, it is natural to expect that the scatterings between the local spins and phonons will be reduced as the magnetization approaches saturation. In fact, the thermal conductivity of 3 T is quite close to the value of 5 T and 7 T, which coincides with the critical field observed in the magnetization measurements. Second, the thermal conductivities are field-independent between 5 T and 7 T. This indicates that even if the magnetic moments are not fully static in our experiment temperature range, they do not scatter phonons since the field has no effect on the thermal conductivities. This behavior strongly suggests that the high-field thermal conductivities are due solely to phonons without magnetic scatterings. Note that there may remain scatterings from structure disorder like stacking faults and grain boundaries.

**Verification of Wiedemann–Franz Law.** In Fig. 3, we fit the thermal conductivity data below 0.3 K for $\mu_0H = 5$ T to examine the WF law in $Pr_2Ir_2O_7$. At ultra-low temperatures, thermal conductivity usually can be fitted to $\kappa/T = a + bT^{\alpha-1}$, where $aT$ represents electrons and other fermionic quasiparticles such as spinons, while $bT^\alpha$ represents phonons and other bosonic quasiparticles such as magnons[28,29]. For phonons, the power $\alpha$ is typically between 2 and 3, due to the specular reflections at the sample surfaces[28,29]. The fitting gives $\kappa_0/T \equiv a = 0.031 \pm 0.008$ mW $K^{-2}$ $cm^{-1}$ and $\alpha = 2.41 \pm 0.13$ for sample A. From Fig. 1b, $\rho_0(5\,T) = 764$ $\mu\Omega$ cm is estimated, giving the WF law expectation $L_0/\rho_0(5\,T) = 0.032$ mW $K^{-2}$ $cm^{-1}$ with $L_0 = 2.45 \times 10^{-8}$ W $\Omega$ $K^{-2}$. Therefore, the WF law is verified nicely. In order to confirm this result, the data of another sample B are also

**Fig. 2 Heat transport results of $Pr_2Ir_2O_7$. a** The thermal conductivity of $Pr_2Ir_2O_7$ single crystal at various magnetic fields along the [111] direction. At zero field, the thermal conductivity is strongly suppressed above $T_s \approx 0.12$ K. **b** The data of thermal conductivity in (**a**) after subtracting the data of 5 T. A collapse region is clearly shown below the temperature at which the arrows point. This suppression temperature is defined as $T_s$. Inset: field dependence of the suppression temperature $T_s$. The error bars reflect the uncertainty in determining the temperature above which the data deviate from 0. They are estimated as the possible suppression temperature range due to the discrete data points. **c** The magneto-thermal conductivity $MTC = (\kappa(H)-\kappa(0\,T))/\kappa(0\,T) \times 100\%$ at various temperatures. Above 5 T, the thermal conductivity tends to saturate.

T, the thermal conductivity data overlap with each other. With decreasing the field, while $\kappa/T$ data still overlap with the high-field curves below a certain temperature $T_s$, they are suppressed more and more strongly above $T_s$. In Fig. 2b, the collapse region of $\kappa/T$ is shown in detail. The arrows in Fig. 2b denote the suppression temperatures $T_s$ at corresponding fields. $(\kappa(H)-\kappa(5\,T))/T$ start to deviate from 0 above $T_s$. At zero field, $T_s$ is about 0.12 K. Similar behavior is also observed in another sample B (for details, see Supplementary Note 3). The field-dependence of $T_s$ is plotted in the inset of Fig. 2b.

The magneto-thermal conductivity $MTC = \Delta\kappa(H)/\kappa(0\,T) = (\kappa(H) - \kappa(0\,T))/\kappa(0\,T) \times 100\%$ at various temperatures is plotted in Fig. 2c. MTC tends to saturate above 5 T below 0.8 K, when the thermal conductivity curves start to overlap with each other. In contrast to the magnetoresistance of less than 5% in charge

plotted in Fig. 3. The fitting gives $\kappa_0/T = 0.033 \pm 0.006$ mW K$^{-2}$ cm$^{-1}$ and $\alpha = 2.62 \pm 0.12$. Since it has $\rho_0(5\,\text{T}) = 755\,\mu\Omega$ cm, thus $L_0/\rho_0(5\,\text{T}) = 0.032$ mW K$^{-2}$ cm$^{-1}$, the WF law is also verified in sample B. The verification of WF law above $\mu_0 H = 5$ T is reasonable, because the thermal conductivity above $\mu_0 H = 5$ T below 0.8 K is purely contributed from normal electrons and phonons, without other exotic excitations or magnetic scatterings.

Since the thermal conductivity data at low fields collapse on the high-field data below $T_s$ (see Fig. 2b) and the MR is less than 2% for $\mu_0 H \leq 5$ T (see Fig. 1b), it would be inferred that the WF law is obeyed at all the applied fields, even at zero field. This result is of great help to characterize the quantum criticality in $Pr_2Ir_2O_7$. Many efforts have been made to describe the QCP phenomena[30], among which two formalisms are highlighted: The Hertz-Millis formalism[31,32] and the Kondo breakdown formalism[33]. In the former one, the critical fluctuations are centered at a small part of the Fermi surface, called hot spots, leaving the majority unaffected and the electrons retaining as Landau quasiparticles. The WF law will be satisfied in this type of QCP due to the integrity of the electrons. In the latter one, the hot spots cover the whole Fermi surface and the critical fluctuations reconstruct the Fermi surface abruptly. The WF law will be violated in this type of QCP due to the breakdown of quasiparticles. The verification of the WF law in $Pr_2Ir_2O_7$ at its QCP unambiguously excludes the possibility of the breakdown of Landau quasiparticles, and is incompatible with the Kondo breakdown formalism. However, this does not immediately indicate a Hertz–Millis type quantum criticality in $Pr_2Ir_2O_7$, because the scaling exponents in the magnetic Grüneisen ratio experiment differ from the expectations within the Hertz–Millis theory[10]. Therefore, new kinds of quantum criticality where the quasiparticle picture is applicable may be realized in $Pr_2Ir_2O_7$, and the confirmation of the WF law puts strong constraints on the description of such QCP.

**Absence of positive contributions to $\kappa$ from magnetic excitations.** Also, the verification of WF law in $Pr_2Ir_2O_7$ demonstrates that there is no additional contribution to the thermal conductivity from mobile fermionic magnetic excitations. Furthermore, since the phonon thermal conductivity in high fields defines the upper boundary of $\kappa$ in $Pr_2Ir_2O_7$, there is also no positive contribution to $\kappa$ from other bosonic magnetic excitations. For pyrochlore $Pr_2Ir_2O_7$, one scenario to describe its possible spin liquid state is the QSI (refs. [3,13]). Three topological excitations, including photon, vison, and magnetic monopole, may emerge from the ground state[3]. The gapless photons have a rather narrow bandwidth, about 1/1000 of the nearest-neighbor coupling $J_{zz}$ (ref. [34]). Since $J_{zz}$ of $Pr_2Ir_2O_7$ is only 1.4 K (ref. [8]), the photons are definitely beyond the accessible temperature regime of our experiment. The gap of visons is about $J_{\perp}^3/J_{zz}^2$, in which $J_{\perp}$ is the transverse exchange coupling between local moments[3]. Taking $J_{\perp}/J_{zz} = 0.3$, a typical value for QSI materials[34–36], the vison gap is estimated as 40 mK for $Pr_2Ir_2O_7$, which is also likely beyond the experiment temperature range. For magnetic monopoles, however, with the gap comparable to $J_{zz}$, the thermally excited magnetic monopoles should be detectable in our temperature range. But we do not observe their positive contributions here. One possibility is that the velocity and/or mean free path of these excitations may be too small so that their contribution to $\kappa$ is negligible comparing to that of phonons. Another possibility is that the ground state of $Pr_2Ir_2O_7$ is not a QSI, thus the above-mentioned magnetic excitations do not exist.

**Giant isotropic MTC and its origin.** Now let us discuss the origin of the giant MTC in $Pr_2Ir_2O_7$. In Fig. 2a, starting from the phonon thermal conductivity in high fields, the strong

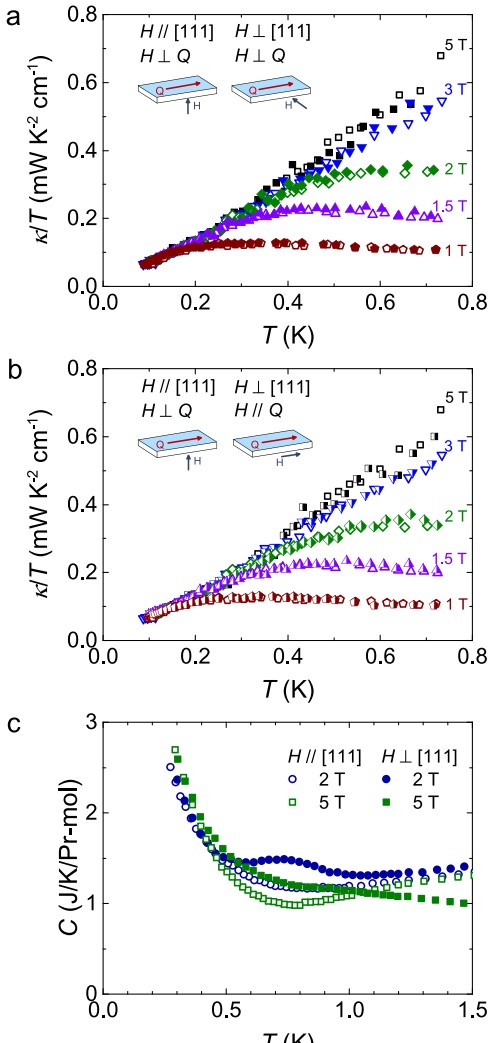

**Fig. 4 The giant isotropic magneto-thermal conductivity and anisotropic magneto-specific heat in $Pr_2Ir_2O_7$. a** The thermal conductivities of $Pr_2Ir_2O_7$ single crystal in magnetic fields $H \parallel$ [111] and $H \perp Q$ (empty symbols) are compared with those in $H \perp$ [111] and $H \perp Q$ (filled symbols), where $Q$ is the heat current. **b** They are also compared with those in $H \perp$ [111] and $H \parallel Q$ (half-filled symbols). The schematic illustrations of these three geometries are also shown. The overlap of these curves in all three field directions show that the MTCs are isotropic. **c** The specific heats of $Pr_2Ir_2O_7$ in magnetic fields $H \parallel$ [111] (empty symbols) are also compared with those in $H \perp$ [111] (filled symbols). Contrary to the magneto-thermal conductivities, anisotropy of magneto-specific heats is clearly seen in these two field directions.

suppression of $\kappa$ at low fields apparently comes from the scattering of phonons by the spin system through the spin-lattice coupling, either by well-defined excitations like magnetic monopoles, which accompany flips of the magnetic dipole moments pointing along the local <111> axes, or by transverse quantum fluctuations, possibly including the quadrupole moments of $Pr^{3+}$. Since the Ir 5d spins are Pauli paramagnetic and polarized due to the Kondo coupling[6,7], the Ir moments are not responsible for the scattering process. In order to examine these possibilities, we measure the thermal conductivity in other two different field directions and compare to the [111] direction, as shown in Fig. 4a and b. One can see that the curves in all three field directions overlap with each other. It shows that the MTC is isotropic, i.e., insensitive to the direction of field.

However, it has been well established that different magnetic structures may emerge in different field directions in the dipole spin-ice formalism[37]. For example, threefold degenerate kagome-ice states can be stabilized for H // [111], while fields along [110] can align the spins on two independent chains. These different magnetic structures induce highly anisotropic specific heat in $Dy_2Ti_2O_7$ (refs. [38,39]). Especially, the different field directions allow the Dy moments to develop phase transitions by decoupling some of the Ising moments[40,41]. For $Pr_2Ir_2O_7$, anisotropic specific heats at $\mu_0 H = 2$ T and 5 T in two field directions are also observed, as shown in Fig. 4c. The low-temperature upturn is attributed to the Schottky anomaly from Pr nuclei. Since specific heats of phonons and electrons are isotropic in response to magnetic fields, the anisotropic part must be contributed from the spin system, likely the dipole moments as in $Dy_2Ti_2O_7$. As a result, the scattering rate between phonons and monopoles $\Gamma_{p\text{-}m}$ should be anisotropic, if one only considers the classical spin ice scenario originating from the local dipole moments. This leads to an anisotropic phonon thermal conductivity $\kappa_p = 1/3 C_p v_p^2/(\Gamma_{p\text{-}m} + \Gamma_{other})$, where $C_p$, $v_p$, $\Gamma_{other}$, are the phonon specific heat, phonon velocity and scattering rate from other mechanism like defects, respectively. Therefore, the isotropic MTC in $Pr_2Ir_2O_7$ indicates that the scattering of phonons is unlikely from magnetic monopoles, the quasiparticles emerging from spin-ice physics due to flips of dipole moments.

Now that the well-defined magnetic excitations such as magnetic monopoles neither contribute to $\kappa$ nor suppress $\kappa$ in $Pr_2Ir_2O_7$, the strong scattering of phonons at low fields may be associated with the transverse quantum fluctuations. $Pr^{3+}$ can be represented by a 1/2-pseudospin. The z component along the local <111> direction carries a magnetic dipole moment, while the transverse component of the pseudospin corresponds to a quadrupole moment. Transverse fluctuations of moments away from the quantization axis, characteristics of the quantum spin ice, can induce quantum dynamics, and may play an important role in this system. Intuitively, the transverse fluctuations are weakened with lowering the temperature and increasing the field. For $Pr_2Ir_2O_7$, a bifurcation of the field-cooled and zero-field cooled magnetic susceptibility curves at about 0.12 K suggests that a partial fraction of spins freezes[6,8], which coincides with $T_s$ of zero field in our thermal conductivity measurements. Therefore, this scenario may explain the temperature and field dependence of thermal conductivity in $Pr_2Ir_2O_7$. In each field, the fluctuations are very slow below $T_s$, due to the partially frozen spins, so that they do not scatter phonons. The field will further weaken the phonon scatterings, thus the $T_s$ increases with increasing field. Such a simple scenario was used to interpret the thermal conductivity of $YbMgGaO_4$ (ref. [27]). One may consider whether it can also apply to the heat transport behavior of other QSI candidates such as $Pr_2Zr_2O_7$ (ref. [36]), as recently pointed out by Rau and Gingras in ref. [42]. Nevertheless, unlike the anisotropic fluctuations found in other spin ice materials, the fluctuations in $Pr_2Ir_2O_7$ should be isotropic in response to the applied field.

## Discussion

As a topical system, thermal conductivities have been measured in various spin ice materials[36,43–45]. Highly anisotropic MTCs were discovered both in $Dy_2Ti_2O_7$ and in another QSI candidate with non-Kramers doublets $Tb_2Ti_2O_7$ (refs. [43,44]). The anisotropic MTC in $Dy_2Ti_2O_7$ was considered as a consequence of the mobility of magnetic monopole excitations in spin ice[43]. The anisotropic MTC with respect to H // [111] and H ⊥ [111] in $Tb_2Ti_2O_7$ was interpreted as a result of the scattering with phonons by anisotropic fluctuating spins[44]. These behaviors are quite contrary to the isotropic MTC in $Pr_2Ir_2O_7$. The isotropic MTC in

$Pr_2Ir_2O_7$ indicates that it is not fluctuations from Ising dipole moments but fluctuations from transverse part, likely quadrupole moments, that scatter phonons strongly above $T_s$. This implication is important since the transverse interactions in addition to the classical Ising interactions is a key ingredient to drive the classical spin ice state into the $U(1)$ QSI state[3]. For a simplest example, when introducing a transverse coupling $J_\perp$ into a classical spin ice model, the resulting Hamiltonian

$$H = \sum_{<i,j>} \left[ J_{zz} \tau_i^z \tau_j^z - J_\perp \left( \tau_i^+ \tau_j^- + \tau_i^- \tau_j^+ \right) \right] \qquad (1)$$

can result in quantum fluctuations and capture the universal properties of $U(1)$ QSI states, where pseudospin-1/2 component $\tau^z$ is along the local <111> direction carrying a magnetic dipole moment, and $\tau^{x,y}$ represent the quadrupole moments giving $\tau^\pm = \tau^x \pm i\tau^y$ (ref. [3]). The possible observation of the quadrupole moment scatterings in $Pr_2Ir_2O_7$ suggests the presence of quantum fluctuations, which is a characteristic of QSI state. Indeed, due to the relatively small magnitude of the $Pr^{3+}$ moments which reduces the dipolar interactions, quadrupolar interactions are expected to be important in Pr-based pyrochlore compounds, and quantum fluctuations therein can melt classical spin ice states[46]. This has been experimentally evidenced both in $Pr_2Sn_2O_7$ and $Pr_2Zr_2O_7$ (refs. [47,48]). To our knowledge, the roles of quadrupole moments and quantum fluctuations in $Pr_2Ir_2O_7$ have not been explored experimentally. Our results present the evidence of quantum effect in this spin liquid candidate, and put its $4f$ moments in line with those in other Pr-based QSI candidates.

The magnetic excitation is another characteristic besides the quantum fluctuation that determines whether a material lies in the QSI ground state. It has been claimed that the thermally excited magnetic monopoles contribute to the thermal conductivity in the QSI candidate $Yb_2Ti_2O_7$, with the mean free path of such monopoles as long as 100 nm (ref. [45]). More exotically, a two-gap behavior and an enhancement below 200 mK are observed in the thermal conductivity of another QSI candidate $Pr_2Zr_2O_7$, which are suggested as the signatures of the three emergent excitations predicted in QSI materials[36]. However, here in $Pr_2Ir_2O_7$, neither positive contributions nor negative contributions to $\kappa$ from well-defined magnetic excitations are detected. This may be a deviation from the QSI scenario despite the presence of quantum fluctuations. However, we should note that although no contributions from magnetic excitations are detected in another QSI candidate $Tb_2Ti_2O_7$ by the longitudinal heat transport study[44], large thermal Hall effect is observed and indicates the existence of possible neutral spin excitations[49]. Therefore, thermal Hall measurements would be intriguing as a tool to determine the true ground state and emergent excitations in $Pr_2Ir_2O_7$.

The above discussion is restricted in the dipole spin ice or QSI categories. However, due to the unique metallic nature of $Pr_2Ir_2O_7$, which is distinct from any other insulating spin ice candidates, the interactions between Pr $4f$ moments and Ir $5d$ itinerant electrons do complicate the microscopic description and induce further correlations besides the spin–ice correlations[6–10]. When considering the effect of conduction d electrons, another possible origin of giant isotropic MTC may be the Kondo coupling between conduction electrons and local moments. The complex $f$-$d$ coupling between Pr moments and Ir electrons may well give rise to a magneto-thermal response which is itself isotropic. The Kondo coupling is also suggested as the ingredient to produce the divergence in the magnetic Grüneisen ratio measurements[50,51], which is also independent of field directions. Interestingly, a scaling behavior of phonon thermal conductivities is observed, similar to the magnetic Grüneisen ratio[10] (for details, see Supplementary Note 2). To check whether the $f$-$d$ interactions

result in the isotropic MTC, it is meaningful to measure the field orientation dependence of $\kappa$ in $Pr_2Zr_2O_7$, where both quantum fluctuations and three exotic magnetic excitations were claimed to be observed without the contamination of conduction electrons[36,48]. Our results raise the question how interactions between the local moment system of a spin-ice state and conduction electrons of a non-Fermi liquid state affect the heat transport properties and the ground state of $Pr_2Ir_2O_7$.

In summary, we have measured the ultralow-temperature thermal conductivity of $Pr_2Ir_2O_7$ single crystals. The Wiedemann–Franz law is verified at high magnetic fields and inferred at zero field, suggesting the normal behavior of electrons at the zero-field quantum critical point and the absence of mobile fermionic magnetic excitations. This result puts strong constraints on the description of the quantum criticality in $Pr_2Ir_2O_7$. A giant isotropic magneto-thermal conductivity is found at finite temperatures, indicating that the quadrupolar interactions and quantum fluctuations may play important roles. Although further experimental work is required to investigate $Pr_2Ir_2O_7$ comprehensively, our results suggest the importance of the quantum effect that deviates from the dipole spin-ice physics, and shed light on the future theoretical and experimental studies on determining the true ground state and magnetic excitations in $Pr_2Ir_2O_7$.

## Methods

**Sample preparation.** We prepare the high-quality single crystals of $Pr_2Ir_2O_7$ using the KF flux methods, as described in ref. [14]. A polycrystalline sample of $Pr_2Ir_2O_7$ was first prepared by the solid-state reaction using stoichiometric starting materials $Pr_2O_3(99.9\%)$ and $IrO_2(99.9\%)$. Then, single crystals were grown by combing the polycrystalline $Pr_2Ir_2O_7$ with potassium fluoride (KF) flux in a ratio of 1:200. After being heated at 1100 °C for 36 h, the as-grown sample was slowly cooled down to 800 °C at a rate of 1 °C/h. The single crystals were obtained from the flux by dissolving the flux in water. The x-ray diffraction (XRD) measurement was performed on a typical $Pr_2Ir_2O_7$ sample by using an x-ray diffractometer (D8 Advance, Bruker), and determined the largest surface to be the (111) plane (see Supplementary Note 1).

**Specific heat measurements.** The low-temperature specific heat was measured by the relaxation method in a physical property measurement system (PPMS, Quantum Design) equipped with a dilution refrigerator.

**Charge and heat transport measurements.** One $Pr_2Ir_2O_7$ single crystal (sample A) for electric and thermal conductivity measurements was cut and polished into a rectangular shape of dimensions $0.69 \times 0.38$ mm$^2$ in the (111) plane, with a thickness of 0.20 mm. Another $Pr_2Ir_2O_7$ single crystal (Sample B) with the dimension of $0.72 \times 0.62 \times 0.20$ mm$^3$ was also measured to check the reproducibility of the transport results (for details, see Supplementary Note 3). If not specified, the data in the main text were taken on Sample A. The electric resistivities were measured using four-contact geometry in a $^3$He cryostat. The thermal conductivities at sub-Kelvin temperatures were measured in a dilution refrigerator, using a standard four-wire steady-state method with two RuO$_2$ chip thermometers, calibrated in situ against a reference RuO$_2$ thermometer. The thermal conductivities above 5 K were measured in a $^4$He cryostat, using a Chromel-Constantan thermocouple as the thermometer to detect the temperature difference. The electric current and heat current were applied in the (111) plane. The charge and heat transport data of the same sample were collected on the same geometry so that the geometry factor could cancel out when checking the Wiedemann-Franz law.

## Data availability

The data that support the findings of this study are available from the corresponding author upon reasonable request.

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

## Acknowledgements

We thank E.-G. Moon, Z.Y. Meng, Y. Wan, and Y. Zhou for helpful discussions. This work is supported by the Natural Science Foundation of China (Grant No. 12034004 and 11421404), the NSAF (Grant No.: U1630248), the Ministry of Science and Technology of China (Grant No.: 2016YFA0300503), the Shanghai Municipal Science and Technology Major Project (Grant No. 2019SHZDZX01), and the fundamental research funds for central universities (2018KFYYXJJ038).

## Author contributions

S.Y.L. conceived the idea and designed the experiments. J.M.N. performed the specific heat and transport measurements with help from Y.Y.H., E.J.C., Y.J.Y., B.L.P., and Q.L. L. M.X. and Z.M.T synthesized the single crystal samples. J.M.N. and S.Y.L. wrote the manuscript with comments from all authors.

## Competing interests

The authors declare no competing interests.
