## [Peer Review File · Nature Communications]

Reviewers' comments:

Reviewer #1 (Remarks to the Author):

NCOMMS-19-40731

Giant isotropic magneto-thermal conductivity of metallic spin liquid candidate Pr₂Ir₂O₇

J.M. Ni et al.

This paper reports measurements of the thermal conductivity in the enigmatic metallic spin liquid candidate Pr₂Ir₂O₇. This material has been extensively studied for nearly fifteen years and we still lack a good understanding of it. It has been suggested from previous works that the material is at (or close) to a quantum critical point (QCP). This work addresses this hypothesis by measurement the transport properties of this compound. The main results of this paper are (1) that the Wiedemann-Franz law is obeyed in this material even in zero field and that there is no breakdown of the Landau quasiparticles at/near the QCP without new fermionic excitations of magnetic origin present and that (2) there exist a large isotropic thermal conductivity in this system despite the intrinsic anisotropic magnetic response that this material should display at $T \sim 1\text{K}$ from the anisotropy of the Pr magnetic moments. There is also the accompanying claim that all thermal conductivity above the electronic component at $T > \sim 100\text{mK}$ is due to phonons and non-magnetic in origin).

While the raw experimental results are certainly interesting and deserve to be published in some form, I do not feel that the present work advances sufficiently our understanding of Pr₂Ir₂O₇ to warrant publication in Nature Communications and I therefore recommend that it be rejected. I also find that some of the claims made in the paper are way too strong given the available experimental evidence (and/or analysis of the data) at hand. More technical comments follow below.

A) General comments:

1. Foremost, the authors argue that the Wiedemann-Franz (WF) law is obeyed in this system in zero magnetic field with the physical implications stated in the opening paragraph of this report. The main support for this claim is that (i) the extrapolation of κ/T vs T in large magnetic field

(Figures 2, 3 and 4) indicate that the WF Law is obeyed (on the basis of the in-field zero-temperature extrapolation) and that (ii) the zero field κ/T “catches” the (say 5Tesla κ/T value at $T \sim T_s \sim 125\text{mK}$ (Figure 3) and, perhaps, sticks with it from 125mK down to 100mK and would extrapolate to the $\lim_{T \rightarrow 0} \{\kappa(T)/T\}$ value expected on the basis of the WF law. While the in-field extrapolation to $T=0$ appears reasonable, I am not so convinced about the zero field results. Furthermore, in absence of an understanding of the physics governing the change of behavior at $T_s(H=0)$, one does not really know why $\kappa(T)/T$ changes reasonably abruptly at $T_s(H=0) \sim 125\text{mK}$. Perhaps most importantly, is the fact that κ/T is actually almost 3 times the WF value and temperature independent from T_s all the way to 800mK. This seems to me almost a more interesting result, but the authors do not dwell on this as they focus on a quest for fermionic quasiparticles that might exist at the QCP above and beyond the Landau quasiparticles.

2. The authors claim that the temperature dependence of κ/T for large field is due solely to phonons. Unless I missed it, I did not see the explicit evidence for that. To support this, the authors should really provide some evidence that the largest field κ/T results has reached the maximum value compatible with boundary limited phonon thermal conductivity (as was done for other spin ice systems such as $\text{Pr}_2\text{Zr}_2\text{O}_7$ [Ref. 31]; PRL 110, 217209 (2013) and PRB 98, 134446 (2018)). Possibly, the thermal conductivity of the present $\text{Pr}_2\text{Ir}_2\text{O}_7$ samples may be lower than the expected boundary-limited phonon thermal conductivity. This may be due to disorder or the fact that there remains scattering of the phonons by magnetic fluctuations. Furthermore, given the small moment of Pr, I am not sure that even at $H=5\text{T}$, the magnetic moments in the kagome plane, which do not experience the full Zeeman energy for this [111] field geometry, will be fully static at $T \sim 800\text{mK}$. Perhaps measurements with a field along [100] might have been better to guarantee a same degree of Pr polarization on all sublattices.

At any rate, a more careful investigation/demonstration that the thermal conductivity is due solely to phonons should be presented.

3. Given that one of the key context the paper uses to discuss the results is that of (quantum) spin ice, I feel that the results should have been contrasted more systematically in relation to thermal conductivity results on spin ice [Refs. [30,31, 35, 36] and the PRL and PRB references above

4. This may be my lack of understanding, but it seems to me that the authors perspective is all wrapped around the spin ice physics that Pr might have supported if it had ordered. I am a bit confused by this because the other pyrochlore iridates $\text{R}_2\text{Ir}_2\text{O}_7$ all show a metal-insulating transition (whose T_{MIT} evolves smoothly with the 4f R element) and an all-in/all-out ordering of the Ir moments at $T \sim 10^2\text{K}$. What has happen to the magnetic character at the Ir site once having moved to the $\text{Pr}_2\text{Ir}_2\text{O}_7$ and down to $T \sim 1\text{K}$? I believe the perspective of Ref. [11] is relevant to this question, but it does not really seem to come up much in this manuscript. Shouldn't it?

B) More specific comments:

5. What is the compelling evidence that the resistivity minimum at 50K is due to Kondo effect as opposed to other mechanisms?

6. What is the authors' perspective/explanation of the physics governing T_s ?

7.

a) The authors use interchangeably the words: spinons, visons, magnetic monopoles, etc. This is confusing and they should adopt one convention, define it in a footnote and stick to it.

In that regard, the authors state that if there were monopoles at scale J_{zz} , they should have been detected in the current experimental temperature window. Why? If the author's magnetic monopoles are the (magnetic) charges associated to the pseudo spin S^z coarse-grained field (aka spinoff), those magnetic monopoles could freeze out at $T \sim 5K$ [DOI: 10.1038/NMAT3900] which is perhaps the scale of J_{zz} . The dual magnetic charges ("visons") would have a much smaller energy gap. For example, in the simplest version of a quantum spin ice, the spin liquid regime is stabilized when the transverse couple $J_{\perp} \ll 0.05J_{zz}$ (from quantum Monte Carlo results). With an estimate of the energy gap of order of the photon bandwidth $\sim 12(J_{\perp})^3/J_{zz}$, one would have a maximum gap of less than 40mK (using $J_{zz} \sim 5K$). So, unless I have not understood, it's not clear to me that those excitations would be readily observed in the work reported here.

b) Possibly related to the same point, the authors refer in the main text and in their discussion to the "fluctuations of spins". What spins? If they are talking about the fluctuations of the Pr ground doublet states, these are already included as quasiparticles (spinons, dual magnetic charge and photon) at $T < O(1K)$. Are they talking about the spin of the conduction electrons?

Reviewer #2 (Remarks to the Author):

In this paper, the ultralow-temperature thermal conductivity of Pr₂Ir₂O₇ single crystals was investigated. One conclusion is that there are no mobile fermionic magnetic excitations. A giant magneto-thermal conductivity was observed, and it is inferred to be due to the strong scattering of phonons by fluctuating spins. Since it was isotropic in response to the magnetic field directions, it is concluded to be different from spin-ice physics. The data is technically sound. The results are novel. The paper provides strong evidence for the conclusions. The paper is important to scientists in the present field. Therefore, the present paper is worth publishing, but high-temperature thermal conductivity data above 1 K are necessary for further discussion. How about the magneto-thermal conductivity above 1 K? How about the magneto-thermal conductivity around 45 K? There may be a relationship between the giant magneto-thermal conductivity and the resistance minimum due to the Kondo effect.

Reviewer #3 (Remarks to the Author):

Referee Report: Nature Communications "Giant Isotropic magneto-thermal conductivity in " by Ni et al.

This paper reports on a study of the thermal conductivity in a magnetic field in the pyrochlore system Pr₂Zr₂O₇, along with associated measurements of the electrical conductivity and the heat capacity. This system is known as a member of the RE₂Ir₂O₇ family of pyrochlore materials which shows a quantum critical-like point as a function of "chemical pressure" as the rare-earth contraction results in smaller lattice constants with increasing atomic number in the rare earth series. Relatively heavy RE members of this family exhibit an insulating ground state with All In - All Out Ir order, and a metallic like ground state for light RE ions such as Nd and Pr.

The authors essentially present two results: one is that the Wiedemann-Franz law appears to be satisfied at relatively high fields in this system, and by extension at lower and presumably zero field; while the other is that the very large magneto-thermal conductivity is ~ isotropic in this system. I feel that the evidence presented for both of these points is reasonably convincing. However, the interpretation is less clear.

The large isotropic magneto-thermal conductivity is unexpected and a striking effect. While I do not consider this to be a seminal development in the field, it is an interesting characteristic of a topical system. I therefore feel that if the authors can adequately address my two specific comments below,

the manuscript would likely be appropriate to Nature Communications. I note that the first of these comments is the more substantive.

The isotropic nature of the very large magneto-thermal conductivity appears to run counter to expectations of a quantum spin ice ground state for the Pr³⁺ moments, as the magnetization for spin ice is known to be anisotropic. However, the exact nature of possible magnetism on the Ir site is not known. In this manuscript, it appears to be assumed that there are no local moments on the Ir site – these would be the same local moments that order into an All-In All Out structure for the heavy rare earth members of the family. I believe all that is known about any possible Ir magnetism in Pr₂Ir₂O₇ is that it is not ordered – fluctuations could still exist. In my opinion the authors should explicitly state that they are assuming there is no magnetism on the Ir site (and give supporting arguments for why this is so), or discuss the possible role of coupled Ir-Pr magnetism. As the Ir magnetic order in the heavy rare earth members of this series is itself non-collinear, the coupling between this and the Pr moments may well give rise to a magneto-thermal response which is itself isotropic. In any case this should be discussed. This is my most significant comment.

The additional comment is somewhat more technical. This is that the resistivity and magnetoresistance shown in Figs 1 a and b and discussed in the manuscript appears to have no information as to any directional dependence (for the resistivity) and any dependence on the magnetic field direction. This seems strange for a paper which is drawing attention to an unexpected isotropic magneto-thermal conductivity. The authors should include this information in the main manuscript.

Prof. Shiyan Li
Laboratory of Advanced Materials
Fudan University
2205 Songhu Road
Shanghai 200438, China
86-21-51630250 (Phone)
86-21-51630249 (Fax)
shiyang_li@fudan.edu.cn

Detailed response to Referees' reports

Referee #1:

This paper reports measurements of the thermal conductivity in the enigmatic metallic spin liquid candidate Pr₂Ir₂O₇. This material has been extensively studied for nearly fifteen years and we still lack a good understanding of it. It has been suggested from previous works that the material is at (or close) to a quantum critical point (QCP) This work addresses this hypothesis by measurement the transport properties of this compound. The main results of this paper are (1) that the Wiedemann-Franz law is obeyed in this material even in zero field and that there is no breakdown of the Landau quasiparticles at/near the QCP without new fermionic excitations of magnetic origin present and that (2) there exist a large isotropic thermal conductivity in this system despite the intrinsic anisotropic magnetic response that this material should display at T<~1K from the anisotropy of the Pr magnetic moments. There is also the accompanying claim that all thermal conductivity above the electronic component at T>~100mK is due to phonons and non-magnetic in origin).

While the raw experimental results are certainly interesting and deserve to be published in some form, I do not feel that the present work advances sufficiently our understanding of Pr₂Ir₂O₇ to warrant publication in Nature Communications and I therefore recommend that it be rejected. I also find that some of the claims made in the paper are way too strong given the available experimental evidence (and/or analysis of the data) at hand. More technical comments follow below.

We thank Referee #1's comments.

A) General comments:

1. Foremost, the authors argue that the Wiedemann-Franz (WF) law is obeyed in this system in zero magnetic field with the physical implications stated in the opening paragraph of this report. The main support for this claim is that (i) the extrapolation of κ/T vs T in large magnetic field (Figures 2, 3 and 4) indicate that the WF Law is obeyed (on the basis of the in-field zero-temperature extrapolation) and that (ii) the zero field κ/T "catches" the (say 5Tesla κ/T value at T<~T_s~125mK (Figure 3) and, perhaps, sticks with it from 125mK down to 100mK and would extrapolate to the $\lim_{T \rightarrow 0} \{\kappa(T)/T\}$ value expected on the basis of the WF law. While the in-field extrapolation to T=0 appears reasonable, I am not so convinced about the zero field results. Furthermore, in absence of an understanding of the physics governing the change of behavior at T_s(H=0), one does not really know why $\kappa(T)/T$ changes reasonably abruptly at T_s(H=0)~125mK. Perhaps most importantly, is

the fact that κ/T is actually almost 3 times the WF value and temperature independent from T_s all the way to 800mK. This seems to me almost a more interesting result, but the authors do not dwell on this as they focus on a quest for fermionic quasiparticles that might exist at the QCP above and beyond the Landau quasiparticles.

We thank Referee #1's comment. We believe that it is reasonable to infer that the Wiedemann-Franz law is obeyed at zero field. The reasons are as follows. Firstly, the overlap between the zero-field data and high-field data below $T_s \sim 120$ mK is obvious and solid, as can be seen from Fig. 2b. More importantly, the overlap can be reproduced in another sample (i.e. sample B in our manuscript). Such solid and reproducible result convinces us that below a certain temperature T_s , the zero-field data indeed overlap with the high-field data, which makes the WF law obeyed at zero field (considering the very small magnetoresistance). Secondly, with decreasing the field from 5 T to 0 T, the overlap of the thermal conductivity data between high-field curves and low-field curves below a certain temperature T_s is **systematic**. In other words, the overlap behavior is observed at each field. Therefore, it is unreasonable to neglect the aforementioned overlap despite the small temperature range, and think that the thermal conductivity is independent of temperature down to 0 K. Last but not least, since the T_s of zero field coincides with the freezing temperature T_f reported in Ref. 6 in our manuscript, we think that perhaps the overlaps of the thermal conductivities come from the partial freezing of spins, since the frozen spins do not scatter phonons. This will be explained in detail in the response to the 6th comment. In this context, we believe that the WF law is verified at zero field, although the alternative scenario may be more fascinating. We hope the Referee #1 is satisfied with our explanation. Thank you!

2. The authors claim that the temperature dependence of κ/T for large field is due solely to phonons. Unless I missed it, I did not see the explicit evidence for that. To support this, the authors should really provide some evidence that the largest field κ/T results has reached the maximum value compatible with boundary limited phonon thermal conductivity (as was done for other spin ice systems such as Pr₂Zr₂O₇ [Ref. 31]; PRL 110, 217209 (2013) and PRB 98, 134446 (2018)). Possibly, the thermal conductivity of the present Pr₂Ir₂O₇ samples may be lower than the expected boundary-limited phonon thermal conductivity. This may be due to disorder or the fact that there remains scattering of the phonons by magnetic fluctuations. Furthermore, given the small moment of Pr, I am not sure that even at H=5T, the magnetic moments in the kagome plane, which do not experience the full Zeeman energy for this [111] field geometry, will be fully static at T~800mk. Perhaps measurements with a field along [100] might have been better to guarantee a same degree of Pr polarization on all sublattices.

At any rate, a more careful investigation/demonstration that the thermal conductivity is due solely to phonons should be presented.

We thank Referee #1's valuable suggestion. We have carefully estimated the mean free path l_p of phonons to investigate whether the boundary scattering limit is achieved at high fields. l_p can be estimated by the kinetic formula $\kappa = 1/3 C_p v_p l_p$, where $C_p = \beta T^3$ is the phonon specific

heat and v_p is the velocity of phonons. With $\beta = 0.766 \text{ mJ mol}^{-1} \text{ K}^{-4}$, taken from the $4f$ moment free compound $\text{Eu}_2\text{Ir}_2\text{O}_7$ as the reference for the phonon specific heat (K. Blacklock, *et al.*, J. Chem. Phys. **72**, 2191 (1980)), v_p is estimated as $2.3 \times 10^4 \text{ m/s}$. Therefore, l_p is estimated as $32 \text{ }\mu\text{m}$ at 0.3 K and 5 T for sample A. When the sample enters the boundary scattering limit, $l_p = 2\sqrt{A/\pi}$, where A is the cross-sectional area of the sample. For sample A, this is about $310 \text{ }\mu\text{m}$. This indicates that the mean free path l_p is $\sim 1/10$ of the value of boundary scattering limit. This is similar to the situation in $\text{Dy}_2\text{Ti}_2\text{O}_7$, where the phonon mean free path is about $1/7$ of the sample size (W. H. Toews, *et al.*, Phys. Rev. B **98**, 13446 (2018)). The structure disorder like stacking faults and grain boundaries or the extra magnetic scatterings may account for the reduced phonon mean free path (W. H. Toews, *et al.*, Phys. Rev. B **98**, 13446 (2018)).

However, we can conclude from the following reasons that the thermal conductivity at 5 T or 7 T is entirely due to phonons with scatterings from structure disorder, rather than from magnetic fluctuations. Firstly, a metamagnetic transition at $B_c \sim 2.3 \text{ T}$ can be induced only when applying the field along the $[111]$ direction in the magnetization $M(B)$ measurements (Ref. 8 in our manuscript). This means that a sizeable fraction of the “2-in, 2-out” configurations are transformed into the “3-in, 1-out” configurations at the critical field. Therefore, it is natural to expect that the scatterings between the local spins and phonons will be reduced as the magnetization approaches to saturation. In fact, the thermal conductivity of 3 T is quite close to the value of 5 T and 7 T , which coincides with the critical field observed in the magnetization measurements. Secondly, the thermal conductivities are field-independent between 5 T and 7 T . This indicates that even if the magnetic moments are not fully static in our experiment temperature range, they do not scatter phonons since the field has no effect on the thermal conductivities. This behavior strongly suggests that the high-field thermal conductivities are due solely to phonons without magnetic scatterings.

On page 4, the third paragraph, we add a paragraph to claim that the thermal conductivity is due solely to phonons without magnetic scatterings more explicitly:

“We would like to emphasize that besides the electron thermal conductivities, κ above 5 T is entirely due to phonons without magnetic scatterings. Firstly, a metamagnetic transition at $B_c \sim 2.3 \text{ T}$ can be induced only when applying the field along the $[111]$ direction in the magnetization $M(B)$ measurements⁸. This means that a sizeable fraction of the “2-in, 2-out” configurations are transformed into the “3-in, 1-out” configurations at the critical field. Therefore, it is natural to expect that the scatterings between the local spins and phonons will be reduced as the magnetization approaches to saturation. In fact, the thermal conductivity of 3 T is quite close to the value of 5 T and 7 T , which coincides with the critical field observed in the magnetization measurements. Secondly, the thermal conductivities are field-independent between 5 T and 7 T . This indicates that even if the magnetic moments are not fully static in our experiment temperature range, they do not scatter phonons since the field has no effect on the thermal conductivities. This behavior strongly suggests that the high-field thermal conductivities are due solely to phonons without magnetic scatterings. Note that there may remain scatterings from structure disorder like stacking faults and grain boundaries.”

Correspondingly, on page 5, the first paragraph, we delete the sentence:

“... because the magnetization approaches saturation⁸ and the system is away from the QCP (ref. 10) at high field. As a result...”

3. Given that one of the key context the paper uses to discuss the results is that of (quantum) spin ice, I feel that the results should have been contrasted more systematically in relation to thermal conductivity results on spin ice [Refs. [30,31, 35, 36] and the PRL and PRB references above

We agree with Referee #1. In the revised manuscript, we add a paragraph in the discussion section to compare these thermal conductivity results more systematically :

“As a topical system, thermal conductivities have been measured in various spin ice materials^{32,38-42}. It has been claimed that the thermally excited magnetic monopoles contribute to the thermal conductivity in the QSI candidate $\text{Yb}_2\text{Ti}_2\text{O}_7$, with the mean free path of such monopoles as long as 100 nm (ref. ³⁸). More exotically, a two-gap behavior and an enhancement below 200 mK are observed in the thermal conductivity of another QSI candidate $\text{Pr}_2\text{Zr}_2\text{O}_7$, which are suggested as the signatures of the three emergent excitations predicted in QSI material³². For classical spin ice materials $\text{Ho}_2\text{Ti}_2\text{O}_7$ and $\text{Dy}_2\text{Ti}_2\text{O}_7$, the magnetic monopoles are reported that they can both conduct heat and scatter phonons^{39,40}. However, here in $\text{Pr}_2\text{Ir}_2\text{O}_7$, neither positive contributions nor negative contributions to κ from magnetic monopoles are detected. Unlike the classical spin ice where high-field thermal conductivity values are basically smaller than the zero-field value³⁹⁻⁴¹, $\text{Pr}_2\text{Ir}_2\text{O}_7$ shows a positive MTC like QSI candidates $\text{Pr}_2\text{Zr}_2\text{O}_7$, $\text{Yb}_2\text{Ti}_2\text{O}_7$, and $\text{Tb}_2\text{Ti}_2\text{O}_7$ (refs. ^{32,38,42}). This might indicate the quantum dynamics of spins in $\text{Pr}_2\text{Ir}_2\text{O}_7$. For the field direction dependence, highly anisotropic MTCs were discovered both in $\text{Dy}_2\text{Ti}_2\text{O}_7$ and in another QSI candidate with non-Kramers doublets $\text{Tb}_2\text{Ti}_2\text{O}_7$ (refs. ^{41,42}). The anisotropic MTC in $\text{Dy}_2\text{Ti}_2\text{O}_7$ was considered as a consequence of the mobility of magnetic monopole excitations in spin ice⁴¹. The anisotropic MTC with respect to $H // [111]$ and $H \perp [111]$ in $\text{Tb}_2\text{Ti}_2\text{O}_7$ was interpreted as a result of the scattering with phonons by anisotropic fluctuating spins⁴². These behaviors are quite contrary to the isotropic MTC in $\text{Pr}_2\text{Ir}_2\text{O}_7$. ”

4. This may be my lack of understanding, but it seems to me that the authors perspective is all wrapped around the spin ice physics that Pr might have supported if it had ordered. I am a bit confused by this because the other pyrochlore iridates $\text{R}_2\text{Ir}_2\text{O}_7$ all show a metal-insulating transition (whose T_{MIT} evolves smoothly with the 4f R element) and an all-in/all-out ordering of the Ir moments at $T < \sim 10^2\text{K}$. What has happen to the magnetic character at the Ir site once having moved to the $\text{Pr}_2\text{Ir}_2\text{O}_7$ and down to $T < \sim 1\text{K}$? I believe the perspective of Ref. [11] is relevant to this question, but it does not really seem to come up much in this manuscript. Shouldn't it?

We thank Referee #1's comment. $\text{Pr}_2\text{Ir}_2\text{O}_7$ is an interesting compound among the pyrochlore iridates $\text{R}_2\text{Ir}_2\text{O}_7$ series with two distinct characteristics: the metallic nature and the absence of magnetic order. The Ir moments in $\text{Pr}_2\text{Ir}_2\text{O}_7$ are different from other antiferromagnetic insulating $\text{R}_2\text{Ir}_2\text{O}_7$ compounds. Firstly, the Ir moments do not order down to low temperatures. Secondly, Ir 5d electrons are conducting in $\text{Pr}_2\text{Ir}_2\text{O}_7$. They are only weakly correlated and Pauli paramagnetic. The contributions to the magnetization $M(B)$ from this Pauli paramagnetic term is several orders smaller than the Pr 4f moments term (Ref. 6 in our manuscript). Thirdly, Ir 5d electrons are subject to the Kondo coupling to Pr 4f moments with the Kondo energy scale of about 20 K. The Pauli paramagnetic Ir spins are polarized due to the fictitious magnetic fields from the spin-chirality effect of Pr moments through the Kondo coupling (Ref. 7 in our manuscript). Given these arguments, the Ir moments are unlikely the source of the strong scatterings of phonons in our thermal conductivity measurements. Just as Ref. 11 points out, the Pr-site ions are important below a few Kelvins in $\text{Pr}_2\text{Ir}_2\text{O}_7$. Therefore, we believe that the Pr magnetic moments are responsible for most of the magnetic properties in this material, and constrain our analysis on the Pr local moments.

On page 7, the second paragraph, to state the role of Ir spins, we add a sentence:

"The fluctuating spins here refer to the Pr 4f spins, because the Ir 5d spins are Pauli paramagnetic and polarized due to the Kondo coupling^{6,7}. Therefore, the Ir moments are not responsible for the scattering process."

B) More specific comments:

5. What is the compelling evidence that the resistivity minimum at 50K is due to Kondo effect as opposed to other mechanisms?

The first evidence of the Kondo effect is the $\ln T$ dependence of the resistivity below 45 K, as can be seen from the inset of Fig. 1a. Note that the crystal electric field (CEF) gap is about 160 K determined by the inelastic neutron scattering measurements (Y. Machida *et al.* J. Phys. Chem. Solids **66**, 1435 (2005)), therefore the $\ln T$ dependence cannot come from the CEF effect, but from Kondo effect. Furthermore, the resistivity can be well fit to the Hamann's equation between 3 K and 35 K (D. R. Hamann, Phys. Rev. **158**, 570(1967)), a quantitative description of Kondo effect. The fitting is now plotted in the inset of Fig. 1a. These resistivity behaviors have been reported in a previous study (Ref. 6 in our manuscript).

In fact, other evidences for Kondo effect have also been reported. For example, the effective Curie constant $C(T) = T\chi(T)$ was found to decrease in the low T , suggesting that the moment size diminishes owing to Kondo screening and resulting a reduced Weiss temperature θ_W (Ref. 6 in our manuscript).

To show the evidence of Kondo effect more explicitly, on page 3, the third paragraph, we rewrite the sentence from:

“The Kondo effect is evidenced by the upturn behavior below 45 K where the resistivity displays a minimum, as shown in the inset of Fig. 1a.”

To

“The upturn behavior and the $\ln T$ dependence below 45 K where the resistivity displays a minimum, and the well fit to the Hamann’s equation are the evidences for Kondo effect, as shown in the inset of Fig. 1a.

Correspondingly, in revised inset of Fig. 1a, the Hamann equation fit is plotted and the caption is also updated.

6. What is the authors’ perspective/explanation of the physics governing T_s ?

The physics governing T_s is equivalent to what makes the thermal conductivities at low fields depart from the pure phonon thermal conductivities above a certain temperature T_s . We think that this strong suppression above T_s must come from the scattering of phonons by the spin system, either by well-defined quasiparticles (magnetic monopoles in this case) or by fluctuating spins. We would like to point out that there are two kinds of spin-phonon scatterings: the inelastic scatterings and elastic scatterings (Ref. 38 in our manuscript). For the former one, it is associated with the spin-flip process, which accompanies a hopping of magnetic monopoles. For the latter one, it is associated with the disorder of spins, which would be suppressed by the alignment of spins or by the slow-dynamics of spins (Ref. 38 in our manuscript).

In the inelastic scattering case that there are thermally excited magnetic monopoles to scatter the phonons above T_s , the field dependence of T_s in Fig. 2(b) suggests that the gap of these magnetic quasiparticles increases with increasing the field. However, the gap of magnetic monopoles should decrease with increasing the field along [111] direction because flipping spins would be easier (Ref. 2 in our manuscript). Furthermore, the scattering between phonons and monopoles should be sensitive to the direction of field in this anisotropic system. But an isotropic MTC is observed in our experiment. Therefore, we believe that the inelastic scattering with quasiparticles is not the physics governing T_s .

Considering the fact that the freezing temperature of 120 mK uncovered by the susceptibility measurements coincides with T_s of zero field in our thermal conductivity measurements (Ref. 6 in our manuscript, note that the freezing temperature of about 300 mK is reported in Ref. 8 in our manuscript), we suggest that the physics governing T_s may be associated with the partial freezing of spins. Note that the elastic scattering process would be suppressed by the alignment of spins or by the slow-dynamics of spins. In this context, in each field, the spins fluctuate very slowly below T_s , partially frozen, so that they do not scatter phonons. In each temperature, with increasing fields, the spins tend to be aligned, resulting in reduced scatterings. Thus, the T_s increases with increasing fields. These are consistent with what we observe in our experiments.

Of course, the behavior of T_s is an interesting phenomenon and the physics governing T_s is an open question. Our speculation and other possible explanations need further study.

7.

a) The authors use interchangeably the words: spinons, visons, magnetic monopoles, etc. This is confusing and they should adopt one convention, define it in a footnote and stick to it.

We thank Referee #1's suggestion. In our manuscript, we adopt the convention that was adopted in the review of quantum spin ice by M. J. P. Gingras and P. A. McClarty (Ref. 3 in our manuscript). In this case, the magnetic monopoles refer to the spin-flip excitations, while the visons refer to the sources of emergent electric fields. The spinons in our manuscript refer to the fermionic fractionalized excitations in other spin liquid systems, such as excitations in the resonant valence bond (RVB) model.

On page2, the first paragraph, we add a sentence to clarify the naming convention of these excitations:

“(In this paper, we adopt the naming convention that the magnetic monopoles refer to the spin-flip excitation, while the visons refer to the sources of emergent electric fields³.)”

In that regard, the authors state that if there were monopoles at scale J_{zz} , they should have been detected in the current experimental temperature window. Why? If the author's magnetic monopoles are the (magnetic) charges associated to the pseudo spin S^z coarse-grained field (aka spinoff), those magnetic monopoles could freeze out at $T \sim 5K$ [DOI: 10.1038/NMAT3900] which is perhaps the scale of J_{zz} . The dual magnetic charges (“visions”) would have a much smaller energy gap. For example, in the simplest version of a quantum spin ice, the spin liquid regime is stabilized when the transverse couple $J_{\perp} \ll -0.05J_{zz}$ (from quantum Monte Carlo results). With an estimate of the energy gap of order of the photon bandwidth $\sim 12(J_{\perp})^3/J_{zz}$, one would have a maximum gap of less than 40mK (using $J_{zz} \sim 5K$). So, unless I have not understood, it's not clear to me that those excitations would be readily observed in the work reported here.

We thank Referee #1's comment. The nearest-neighbor exchange coupling J_{zz} is estimated as 1.4 K by Y. Machida, *et al.* (Ref. 8 in our manuscript). The large-scale quantum Monte Carlo simulation shows that the gapless photons have a rather narrow bandwidth, about 1/1000 of the spin exchange constant J_{zz} (C. J. Huang, *et al.*, Phys. Rev. Lett. **120**, 167202 (2018)). Since J_{zz} of $\text{Pr}_2\text{Ir}_2\text{O}_7$ is only 1.4 K, the photons are definitely beyond the accessible temperature regime of our experiment. The gap of visons is about J_{\perp}^3/J_{zz} , in which J_{\perp} is the transverse exchange coupling between local spins (Ref. 3 in our manuscript). Taking $J_{\perp}/J_{zz} = 0.3$, a typical value for QSI materials (K. A. Ross, *et al.*, Phys. Rev. X **1**, 021002 (2011), Y. Tokiwa, *et al.*, J. Phys. Soc. Jpn. **87**, 064702 (2018), C. J. Huang, *et al.*, Phys. Rev. Lett. **120**, 167202 (2018)), the vison gap is estimated as 40 mK for $\text{Pr}_2\text{Ir}_2\text{O}_7$, which is also likely beyond the experiment temperature range. For magnetic monopoles, however, with the gap

comparable to J_{zz} , the thermally excited magnetic monopoles should be detectable in our temperature range. For example, magnetic monopoles were observed at sub-Kelvin region in the thermal conductivity measurements of the QSI candidate $\text{Yb}_2\text{Ti}_2\text{O}_7$, with J_{zz} of about 2 K (Y. Tokiwa, *et al.*, Nat. Commun. **7**, 10807(2016), K. A. Ross, *et al.*, Phys. Rev. X **1**, 021002 (2011)). Therefore, magnetic monopoles would be observed in our experiments, if they do exist, while other two excitations are not detectable.

On page 6, the first paragraph, to clarify the detectability of these excitations, we rewrite the sentences from:

“Since J of $\text{Pr}_2\text{Ir}_2\text{O}_7$ is only 1.4 K (ref. ⁸), the photons are likely beyond the accessible temperature regime of our experiment. Both visons and magnetic monopoles have a gap³. It has been claimed that the thermally excited magnetic monopoles contribute to the thermal conductivity in the QSI candidates $\text{Yb}_2\text{Ti}_2\text{O}_7$ (ref. ³⁰) and $\text{Pr}_2\text{Zr}_2\text{O}_7$ (ref. ³¹). However, here in $\text{Pr}_2\text{Ir}_2\text{O}_7$, we do not observe their positive contribution. If the magnetic monopoles indeed exist in $\text{Pr}_2\text{Ir}_2\text{O}_7$ with a gap comparable to J , the thermally excited magnetic monopoles should be detectable in our temperature range.”

To:

“Since J_{zz} of $\text{Pr}_2\text{Ir}_2\text{O}_7$ is only 1.4 K (ref. ⁸), the photons are definitely beyond the accessible temperature regime of our experiment. The gap of visons is about $J_{\perp}^{\beta}/J_{zz}^{\beta}$, in which J_{\perp} is the transverse exchange coupling between local spins³. Taking $J_{\perp}/J_{zz} = 0.3$, a typical value for QSI materials³⁰⁻³², the vison gap is estimated as 40 mK for $\text{Pr}_2\text{Ir}_2\text{O}_7$, which is also likely beyond the experiment temperature range. For magnetic monopoles, however, with the gap comparable to J_{zz} , the thermally excited magnetic monopoles should be detectable in our temperature range.”

b) Possibly related to the same point, the authors refer in the main text and in their discussion to the “fluctuations of spins”. What spins? If they are talking about the fluctuations of the Pr ground doublet states, these are already included as quasiparticles (spinons, dual magnetic charge and photon) at $T < O(1\text{K})$. Are they talking about the spin of the conduction electrons?

As has been pointed out in the response to the 6th comment, there are two kinds of spin-phonon scatterings: the inelastic scatterings and elastic scatterings. The inelastic scattering is related to quasiparticles, which we think is not the main reason for the strong suppression of thermal conductivities at low fields above T_s . Indeed, neither positive contributions nor negative contributions to κ from magnetic monopoles are detected in our experiments. One may raise the question whether such quasiparticles indeed exist in $\text{Pr}_2\text{Ir}_2\text{O}_7$. In this sense, we attribute the strong scattering above T_s to the scattering between fluctuating Pr spins and phonons. Note that the Ir magnetism of conduction electrons are negligible compared with the Pr Ising moments, as pointed out in the response to the 4th comment. When increasing the fields, with the alignment of the Pr spins, the fluctuation and disorder of

Pr spins will be reduced, and the elastic scattering will also be suppressed. This is consistent with the positive MTC in our experiment.

Referee #2:

In this paper, the ultralow-temperature thermal conductivity of Pr₂Ir₂O₇ single crystals was investigated. One conclusion is that there are no mobile fermionic magnetic excitations. A giant magneto-thermal conductivity was observed, and it is inferred to be due to the strong scattering of phonons by fluctuating spins. Since it was isotropic in response to the magnetic field directions, it is concluded to be different from spin-ice physics. The data is technically sound. The results are novel. The paper provides strong evidence for the conclusions. The paper is important to scientists in the present field. Therefore, the present paper is worth publishing, but high-temperature thermal conductivity data above 1 K are necessary for further discussion. How about the magneto-thermal conductivity above 1 K? How about the magneto-thermal conductivity around 45 K? There may be a relationship between the giant magneto-thermal conductivity and the resistance minimum due to the Kondo effect.

We appreciate Referee #2's recommendation.

As suggested by Referee #2, we did the thermal conductivity measurements of Sample B between 5 K and 50 K. Due to the limitations of our experimental technique, the temperatures between 1 K and 5 K are unreachable. The results are plotted below and added in the Supplementary materials as Section IV.

The high-temperature thermal conductivities of Sample B are shown in Fig. a. A broad peak is observed at around 10 K at each field. The κ/T data at $\mu_0H = 5$ T still overlap with those at 7 T below 9 K, indicating that besides the electron thermal conductivity, the κ above 5 T is purely contributed by phonons without magnetic scatterings until thermal fluctuations of spins dominate over polarization effects by magnetic fields at even higher temperatures ($T > 9$ K). Unexpectedly, contrary to the ultralow-temperature results, the zero-field data are the largest among the data at other fields and the κ data are suppressed more strongly with increasing the fields. In other words, negative MTCs are observed at temperatures above 5 K, which is opposite to the giant positive MTCs at sub-Kelvin region. Therefore, between 1 K and 5 K, there must exist an exotic crossover from the positive MTCs at low temperatures to the negative MTCs at high temperatures, as shown in the yellow part of Fig. b.

This crossover region is in good correspondence with the energy scale of renormalized interactions between Pr moments, which is $\theta_w = 1.7$ K. When $T < \theta_w$, the susceptibility and Hall resistivity start to exhibit logarithmically diverging behaviors and the anomalous Hall effect emerges (Ref. 6-8 in our manuscript). The magnetic specific heat also exhibits a broad peak at this temperature (Ref. 6 in our manuscript). These all indicate a spin-liquid state below this temperature. This particular energy scale comes from the partial screening of Pr moments due to the Kondo effect, which renormalizes the AFM interaction from the RKKY interaction energy scale of about 20 K to $\theta_w = 1.7$ K. As a result, θ_w is a critical temperature

dividing two separate states: when $T < \theta_w$, the underscreened moments form a correlated spin liquid, where “2-in, 2-out” configurations are the ground states; when $T > \theta_w$, the Kondo effect starts to lead to the screening of the $4f$ moments (Ref. 6,8 in our manuscript). Therefore, the crossover at this temperature region may be due to the different physical states above and below θ_w . In this context, the origin of the negative MTC at high temperatures must be different from the positive MTC at low temperatures, and should be related to the Kondo screening of $4f$ moments. Note that the negative MTC itself is rather unexpected and interesting, and calls for a careful research and analysis in the future.

In our revised manuscript, we add a new section in the Supplementary materials to show and discuss the high-temperature thermal conductivities.

On page 4, the second paragraph, we add a sentence:

“Unexpectedly, a crossover from the positive MTCs at low temperatures to the negative MTCs at high temperatures is observed at θ_w energy scale (for details, see Supplementary materials).”

In the methods part, we add the description of the methods of high-temperature thermal conductivities:

“The thermal conductivities above 5 K were measured in a ^4He cryostat, using a Chromel-Constantan thermocouple as the thermometer to detect the temperature difference.”

Referee #3:

This paper reports on a study of the thermal conductivity in a magnetic field in the pyrochlore system $\text{Pr}_2\text{Zr}_2\text{O}_7$, along with associated measurements of the electrical conductivity and the heat capacity. This system is known as a member of the $\text{RE}_2\text{Ir}_2\text{O}_7$ family of pyrochlore materials which shows a quantum critical-like point as a function of “chemical pressure” as the rare-earth contraction results in smaller lattice constants with increasing atomic number in the rare earth series. Relatively heavy RE members of this family exhibit an insulating ground state with All In - All Out Ir order, and a metallic like ground state for light RE ions such as Nd and Pr.

The authors essentially present two results: one is that the Wiedemann-Franz law appears to be satisfied at relatively high fields in this system, and by extension at lower and presumably zero field; while the other is that the very large magneto-thermal conductivity is ~ isotropic in this system. I feel that the evidence presented for both of these points is reasonably convincing. However, the interpretation is less clear.

The large isotropic magneto-thermal conductivity is unexpected and a striking effect. While I do not consider this to be a seminal development in the field, it is an interesting characteristic of a topical system. I therefore feel that if the authors can adequately address my two specific comments below, the manuscript would likely be appropriate to Nature Communications. I note that the first of these comments is the more substantive.

We appreciate Referee #3's recommendation.

The isotropic nature of the very large magneto-thermal conductivity appears to run counter to expectations of a quantum spin ice ground state for the Pr^{3+} moments, as the magnetization for spin ice is known to be anisotropic. However, the exact nature of possible magnetism on the Ir site is not known. In this manuscript, it appears to be assumed that there are no local moments on the Ir site – these would be the same local moments that order into an All-In All Out structure for the heavy rare earth members of the family. I believe all that is known about any possible Ir magnetism in $\text{Pr}_2\text{Ir}_2\text{O}_7$ is that it is not ordered – fluctuations could still exist. In

my opinion the authors should explicitly state that they are assuming there is no magnetism on the Ir site (and give supporting arguments for why this is so), or discuss the possible role of coupled Ir-Pr magnetism. As the Ir magnetic order in the heavy rare earth members of this series is itself non-collinear, the coupling between this and the Pr moments may well give rise to a magneto-thermal response which is itself isotropic. In any case this should be discussed. This is my most significant comment.

We thank Referee #3's comment. $\text{Pr}_2\text{Ir}_2\text{O}_7$ is an interesting compound among the pyrochlore iridates $\text{R}_2\text{Ir}_2\text{O}_7$ series with two distinct characteristics: the metallic nature and the absence of magnetic order. The Ir moments are different from other antiferromagnetic insulating $\text{R}_2\text{Ir}_2\text{O}_7$ compounds. Firstly, the Ir moments do not order down to low temperatures. Secondly, Ir 5d electrons are conducting in $\text{Pr}_2\text{Ir}_2\text{O}_7$. They are only weakly correlated and Pauli paramagnetic. The contributions to the magnetization $M(B)$ from this Pauli paramagnetic term is several orders smaller than the Pr 4f moments term (Ref. 6 in our manuscript). Thirdly, Ir 5d electrons are subject to the Kondo coupling to Pr 4f moments because of the Kondo energy scale of about 20 K. The Pauli paramagnetic Ir spins are polarized due to the fictitious magnetic fields from the spin-chirality effect of Pr moments through the Kondo coupling (Ref. 7 in our manuscript). Given these arguments, the Ir moments are unlikely the source of the strong scatterings in our thermal conductivity measurements. Therefore, although there still exists Ir magnetism, it has nothing to do with the scatterings between spins and phonons. However, as for the isotropic MTC, the complex *f-d* interactions are indeed a possible explanation, just as pointed out by Referee #3.

On page 7, the second paragraph, to state the role of Ir magnetism, we add a sentence:

“The fluctuating spins here refer to the Pr 4f spins, because the Ir 5d spins are Pauli paramagnetic and polarized due to the Kondo coupling^{6,7}. Therefore, the Ir moments are not responsible for the scattering process.”

On page 8, the second paragraph, we add a sentence:

“The complex *f-d* coupling between Pr moments and Ir electrons may well give rise to a magneto-thermal response which is itself isotropic.”

The additional comment is somewhat more technical. This is that the resistivity and magnetoresistance shown in Figs 1 a and b and discussed in the manuscript appears to have no information as to any directional dependence (for the resistivity) and any dependence on the magnetic field direction. This seems strange for a paper which is drawing attention to an unexpected isotropic magneto-thermal conductivity. The authors should include this information in the main manuscript.

We thank Referee #3's comment. No anisotropy of resistivity was reported with respect to the electric current direction (Ref. 6 in our manuscript). Therefore, for the resistivity measurements, the electric current is applied in the (111) plane. For the magnetoresistance

measurements, the magnetic field is applied along the [111] direction. It was reported that the magnetoresistance exhibits anisotropic response with respect to the field directions, as can be seen in the figure below (Ref. 9 in our manuscript). This was interpreted as a result of the different field-induced magnetic structures, which is quite similar to the specific heats in our experiments and again shows the isotropic MTC is a striking result.

On page3, the third paragraph, to include the information of the directional dependence for the resistivity and the field direction dependence for the magnetoresistance, we add a sentence:

“Note that no anisotropy of resistivity is reported with respect to the electric current direction⁶, while the magnetoresistance is anisotropic in different magnetic field directions⁹.”

In the caption of Fig. 1b, we add a sentence: “The magnetic field is applied along the [111] direction.”

List of changes to manuscript

1) Page2, the first paragraph:

We add a sentence to clarify the naming convention of these excitations:

“(In this paper, we adopt the naming convention that the magnetic monopoles refer to the spin-flip excitation, while the visons refer to the sources of emergent electric fields³.)”

2) Page 3, the third paragraph:

We rewrite a sentence to show the evidence of Kondo effect more explicitly:

“The upturn behavior and the $\ln T$ dependence below 45 K where the resistivity displays a minimum, and the well fit to the Hamann’s equation are the evidences for Kondo effect, as shown in the inset of Fig. 1a.”

3) Page 3, the third paragraph:

We add a sentence to include the information of the directional dependence for the resistivity and the field direction dependence for the magnetoresistance:

“Note that no anisotropy of resistivity is reported with respect to the electric current direction⁶, while the magnetoresistance is anisotropic in different magnetic field directions⁹.”

4) Page 4, the second paragraph:

We add a sentence: “Unexpectedly, a crossover from the positive MTCs at low temperatures to the negative MTCs at high temperatures is observed at θ_w energy scale (for details, see Supplementary materials).”

5) Page 4, the third paragraph:

We add a paragraph to claim that the thermal conductivity is due solely to phonons more explicitly:

“We would like to emphasize that besides the electron thermal conductivities, k above 5 T is entirely due to phonons without magnetic scatterings. Firstly, a metamagnetic transition at $B_c \sim 2.3$ T can be induced only when applying the field along the [111] direction in the magnetization $M(B)$ measurements⁸. This means that a sizeable fraction of the “2-in, 2-out” configurations are transformed into the “3-in, 1-out” configurations at the critical field. Therefore, it is natural to expect that the scatterings between the local spins and phonons will be reduced as the magnetization approaches to saturation. In fact, the thermal conductivity of 3 T is quite close to the value of 5 T and 7 T, which coincides with the critical field observed in the magnetization measurements. Secondly, the thermal conductivities are field-independent between 5 T and 7 T. This indicates that even if the magnetic moments are not fully static in our experiment temperature range, they do not scatter phonons since the field has no effect on the thermal conductivities. This behavior strongly suggests that the high-field thermal conductivities are due solely to phonons without magnetic scatterings. Note that there may remain scatterings from structure disorder like stacking faults and grain boundaries.”

6) Page 5, the first paragraph:

We delete the sentence: “... because the magnetization approaches saturation⁸ and the system is away from the QCP (ref. 10) at high field. As a result...”

7) Page 6, the first paragraph:

We rewrite the sentences to clarify the detectability of these excitations :

“Since J_{zz} of $\text{Pr}_2\text{Ir}_2\text{O}_7$ is only 1.4 K (ref. ⁸), the photons are definitely beyond the accessible temperature regime of our experiment. The gap of visons is about $J_{\perp}^{\beta}/J_{zz}^{\beta}$, in which J_{\perp} is the transverse exchange coupling between local spins³. Taking $J_{\perp}/J_{zz} = 0.3$, a typical value for QSI materials³⁰⁻³², the vison gap is estimated as 40 mK for $\text{Pr}_2\text{Ir}_2\text{O}_7$, which is also likely beyond the experiment temperature range. For magnetic monopoles, however, with the gap comparable to J_{zz} , the thermally excited magnetic monopoles should be detectable in our temperature range.”

8) Page 7, the second paragraph:

On page 7, the second paragraph, to state the role of Ir magnetism, we add a sentence:

“The fluctuating spins here refer to the Pr 4*f* spins, because the Ir 5*d* spins are Pauli paramagnetic and polarized due to the Kondo coupling^{6,7}. Therefore, the Ir moments are not responsible for the scattering process.”

9) Page 7, the third paragraph:

We add a paragraph to compare other spin ice materials' thermal conductivity results more systematically :

“As a topical system, thermal conductivities have been measured in various spin ice materials^{32,38-42}. It has been claimed that the thermally excited magnetic monopoles contribute to the thermal conductivity in the QSI candidate $\text{Yb}_2\text{Ti}_2\text{O}_7$, with the mean free path of such monopoles as long as 100 nm (ref. ³⁸). More exotically, a two-gap behavior and an enhancement below 200 mK are observed in the thermal conductivity of another QSI candidate $\text{Pr}_2\text{Zr}_2\text{O}_7$, which are suggested as the signatures of the three emergent excitations predicted in QSI material³². For classical spin ice materials $\text{Ho}_2\text{Ti}_2\text{O}_7$ and $\text{Dy}_2\text{Ti}_2\text{O}_7$, the magnetic monopoles are reported that they can both conduct heat and scatter phonons^{39,40}. However, here in $\text{Pr}_2\text{Ir}_2\text{O}_7$, neither positive contributions nor negative contributions to κ from magnetic monopoles are detected. Unlike the classical spin ice where high-field thermal conductivity values are basically smaller than the zero-field value³⁹⁻⁴¹, $\text{Pr}_2\text{Ir}_2\text{O}_7$ shows a positive MTC like QSI candidates $\text{Pr}_2\text{Zr}_2\text{O}_7$, $\text{Yb}_2\text{Ti}_2\text{O}_7$, and $\text{Tb}_2\text{Ti}_2\text{O}_7$ (refs. ^{32,38,42}). This might indicate the quantum dynamics of spins in $\text{Pr}_2\text{Ir}_2\text{O}_7$. For the field direction dependence, highly anisotropic MTCs were discovered both in $\text{Dy}_2\text{Ti}_2\text{O}_7$ and in another QSI candidate with non-Kramers doublets $\text{Tb}_2\text{Ti}_2\text{O}_7$ (refs. ^{41,42}). The anisotropic MTC in $\text{Dy}_2\text{Ti}_2\text{O}_7$ was considered as a consequence of the mobility of magnetic monopole excitations in spin ice⁴¹. The anisotropic MTC with respect to $H // [111]$ and $H \perp [111]$ in $\text{Tb}_2\text{Ti}_2\text{O}_7$ was interpreted as a result of the scattering with phonons by anisotropic fluctuating spins⁴². These behaviors are quite contrary to the isotropic MTC in $\text{Pr}_2\text{Ir}_2\text{O}_7$.”

10) Page 8, the third paragraph:

We add a sentence: “The complex f - d coupling between Pr moments and Ir electrons may well give rise to a magneto-thermal response which is itself isotropic.”

11) Page 9, the methods part:

We add the description of the methods of high-temperature thermal conductivities: “The thermal conductivities above 5 K were measured in a ^4He cryostat, using a Chromel-Constantan thermocouple as the thermometer to detect the temperature difference.”

12) Revised Fig. 1a:

In the inset of Fig. 1a, the Hamann equation fit is plotted. The caption is revised accordingly.

13) Supplementary materials:

We add a new section in the Supplementary materials to show and discuss the high-temperature thermal conductivities.

14) Four references are added:

30. Huang, C. J., Deng, Y., Wan, Y. & Meng, Z. Y. Dynamics of topological excitations in a model quantum spin ice, *Phys. Rev. Lett.* **120**, 167202 (2018).

31. Ross, K. A., Savary, L. Gaulin, B. D. & Balents, L. Quantum Excitations in Quantum Spin Ice, *Phys. Rev. X* **1**, 021002 (2011).

39. Toews, W. H., Zhang, S. S., Ross, K. A., Dabkowska, H. A., Gaulin, B. D. & Hill, R. W. Thermal Conductivity of $\text{Ho}_2\text{Ti}_2\text{O}_7$ along the [111] Direction, *Phys. Rev. Lett.* **110**, 217209 (2013).

40. Toews, W. H., Reid, J. A., Nadas, R. B., Rahemtulla, A., Kycia, S., Munsie, T. J. S., Dabkowska, H. A., Gaulin, B. D. & Hill, R. W. Disorder dependence of monopole dynamics in $\text{Dy}_2\text{Ti}_2\text{O}_7$ probed via thermal transport measurements, *Phys. Rev. B* **98**, 134446 (2018).

REVIEWER COMMENTS

Reviewer #1 (Remarks to the Author):

NCOMMS-19-40731

Giant isotropic magneto-thermal conductivity of metallic spin liquid candidate Pr₂Ir₂O₇

J.M. Ni et al.

This is my second report on this manuscript. While the authors have worked hard at addressing the referees' report, including mine, I remain unconvinced that the present work advances sufficiently our understanding of Pr₂Ir₂O₇ to warrant publication in Nature Communications and I, again, recommend that it be rejected.

I do not want to reiterate what I wrote before. The main results of the paper are

(1) a reasonable demonstration of the Wiedemann-Franz (WF) being obeyed at all fields (including zero field) below 0.8K (I say "reasonable" because no data are presented below ~0.1K).

(2) an isotropic magneto-thermal conductivity (MTC).

In regards to (1), the author argue this is an important result because it shows that Landau quasiparticles exist despite the experimental evidence of a quantum critical point (QCP) as revealed by the diverging (and scaling) behavior of the Gruneisen parameter in Ref. [10].

First of all, I do not feel that there has been an exceptionally large amount of work demonstrating the evidence of a QCP in Pr₂Ir₂O₇ and one may question whether the last word has been said about that. If so, then the fulfillment of the WF law would be of lesser importance I would think. That being said, going along assuming that there is indeed a QCP, I then do not feel that the authors really discuss much the significance, importance, consequences of the confirmation of the WF law and what that tells us about the QCP in this system with, at the end, merely report that they find it being satisfied.

In regards to (2): while the observation isotropic MTC is an interesting result (in view of the anisotropic low-temperature specific heat, for example), the authors really do not provide any real

hints as to why it might be so and what is the significance/importance of this result (except, perhaps the statement that the physics of Pr₂Ir₂O₇ deviates from that of spin ice).

While those results are interesting, I find that the authors' inability to provide a springboard as to what and how to think about their two results makes the case for Nature Communications weak. In short, their story for results (1) and (2) just does not go far enough.

I have some specific comments that the authors might want to consider:

- The authors devote quite a bit of space to talk about organized structures in various field directions for spin ice. This is

particularly important for the meta magnetic transition in [111] field. I think that the key point here is that at $T \sim 2\text{K}$, there are

2-in/2-out \gg magnetic \ll correlations in place that provide a "holding" field that has to be overcome for the transition.

I am not sure how far one can assign spin ice physics in the broadest sense of all its phenomenology to Pr₂Ir₂O₇.

My point is that, perhaps, the label of (quantum) spin ice for Pr₂Ir₂O₇ may be used with some caution.

- The authors employ the word "spin" in numerous places in their manuscript: in terms of the spin correlations, spin fluctuations, etc.

I find that their usage of the word spin is possibly confusing  what do they mean by "spin"?

Do they mean the magnetic (dipole) moment, which should be strictly having a local [111] component for

Pr³⁺ being non-Kramers? Or do they refer to the $S=1/2$ pseudospin? The former would be fine while the latter would be confusing

since the transverse parts of the "spin" represent the quadrupole of Pr³⁺.

In this context, it seems that the authors discuss “fluctuations of the spins” and “monopoles” as two separate types of fluctuations (which would be fine in some broad sense if they had in mind the $S=1/2$ pseudospin description, but in other places, they seem to equate spin to dipole moment. I had commented on that in my previous report, and I still find the authors language unclear.

- In lines [160-162], and [210-212], the authors mention the role of monopoles on the thermal conductivity of Dy₂Ti₂O₇ spin ice, but I was unable to grasp what is the point they wish to make about that topic and thermal conductivity (due to monopoles) in Pr₂Ir₂O₇.

- In line #145, the authors write that there is no breakdown of the Landau quasi-particles ... in this non-Fermi liquid. I do not understand how one has fermionic Landau quasi-particles (leading to a satisfied WF Law) but the system being a non-Fermi liquid.

- The authors discuss the anisotropic nature of the specific heat while the MTC is isotropic. I found that the authors do not adequately discuss the link (really, paradox) of having anisotropic C_v but isotropic MTC. In the same vein, the discussion in lines [174-178] could have been clearer. In particular, for Dy₂Ti₂O₇, the different field directions allow the Dy moments to develop phase transitions by decoupling some of the Ising moments (as originally proposed in Nature 399, 333 [1999] and explained in Phys. Rev. Lett. 95, 097202 [2015]).

- Unless I missed it, I could not find what is the field direction for the thermal conductivity measurement with $H \perp [111]$?

Is it $[1 -1 0]$ or some other direction?

- In line [216], the authors write “this might indicate the quantum dynamics of spins in Pr₂Ir₂O₇”. Here, again, the word spin is used and it is not clear at all what is the point/logic the authors aim to make by this statement.

Generally speaking, I find the whole discussion section pretty much void of useful information that might give a possible/plausible suggestion as to what may be going on with the isotropic MTC in Pr₂Ir₂O₇ — It’s pretty much 1.5 page of text that does not say much (including lines 247-250] and, in my opinion, this is why this paper should not be published in Nature Communications as, to no fault to the authors, they really don’t seem to have an idea as to why the MTC is isotropic.

If the MTC came from the electrons (if they were not proper Landau quasiparticles), then one might think the MTC would “have more chance” to be more isotropic than a “spin/lattice” coupling channel. But the authors have ruled this out in the first part of their results section. In that vein, unless I missed it, the authors do not seem to tie together their WF results with their MTC results. I provided one such connection here, but maybe they see others (and better ones). If so, they should mention them.

- Minor comment: what is the meaning of the subscript “s” in T_s ?

Reviewer #2 (Remarks to the Author):

In this paper, the ultralow-temperature thermal conductivity of Pr₂Ir₂O₇ single crystals was investigated. The present paper is worth publishing, but I asked high-temperature thermal conductivity data above 1 K. According to my suggestion, the authors have carried out the magneto-thermal conductivity measurements at high temperatures up to 50 K and discussed the data. Therefore, the revised paper can be published.

Reviewer #3 (Remarks to the Author):

I have read the re-submitted manuscript as well as the authors' response to both my comments and those of the other two referees. My original review of the manuscript was reasonably favourable, but I required commentary regarding the possible nature of the Ir magnetism in generating the isotropic response that is reported. I felt that the authors provided a reasonable response to my point. The most detailed comments appear to come from Ref. 1 - again at the level that I can tell, their response seems to be reasonable. For these reasons I now recommend acceptance as a Nature Communications.

Prof. Shiyan Li
Laboratory of Advanced Materials
Fudan University
2205 Songhu Road
Shanghai 200438, China
86-21-51630250 (Phone)
86-21-51630249 (Fax)
shiyan_li@fudan.edu.cn

Detailed response to Referees' reports

Referee #1:

This is my second report on this manuscript. While the authors have worked hard at addressing the referees' report, including mine, I remain unconvinced that the present work advances sufficiently our understanding of Pr₂Ir₂O₇ to warrant publication in Nature Communications and I, again, recommend that it be rejected.

I do not want to reiterate what I wrote before. The main results of the paper are (1) a reasonable demonstration of the Wiedemann-Franz (WF) being obeyed at all fields (including zero field) below 0.8K (I say "reasonable" because no data are presented below ~0.1K).
(2) an isotropic magneto-thermal conductivity (MTC).

We thank Referee #1 for his/her time reviewing our work and giving valuable suggestions. Please see below our response.

In regards to (1), the author argue this is an important result because it shows that Landau quasiparticles exist despite the experimental evidence of a quantum critical point (QCP) as revealed by the diverging (and scaling) behavior of the Grüneisen parameter in Ref. [10]. First of all, I do not feel that there has been an exceptionally large amount of work demonstrating the evidence of a QCP in Pr₂Ir₂O₇ and one may question whether the last word has been said about that. If so, then the fulfillment of the WF law would be of lesser importance I would think. That being said, going along assuming that there is indeed a QCP, I then do not feel that the authors really discuss much the significance, importance, consequences of the confirmation of the WF law and what that tells us about the QCP in this system with, at the end, merely report that they find it being satisfied.

Besides the diverging and scaling behavior of magnetic Grüneisen ratio measurements (Nat. Mater. 13, 356-359 (2014), Ref. 10 in our manuscript), there are other experiments indicating that Pr₂Ir₂O₇ might be located at a QCP. For example, the low-temperature magnetic susceptibility is found to diverge as $\ln T$ temperature dependence (Phys. Rev. Lett. 96, 087204 (2006), Ref. 6 in our manuscript), which is a hallmark of QCP. Furthermore, an investigation on slight Pr-rich samples Pr_{2+x}Ir_{2-x}O_{7-δ} discovers an antiferromagnetic long-range order at about 0.9 K (D. E. MacLaughlin, *et al.* Phys. Rev. B **92**, 054432 (2015)), suggesting that the stoichiometric samples are located close to a phase transition between a magnetic order state and a disordered state. Therefore, it is reasonable to consider Pr₂Ir₂O₇ as a material located near a QCP.

We thank Referee #1 for pointing out that we should stress more on the importance and implications of the WF law being satisfied. In fact, determining whether quasiparticles survive near the QCP by testing the WF law is of great help to characterize the quantum criticality. Many efforts have been made to describe the QCP phenomena, among which two formalisms are highlighted: The Hertz-Millis formalism and the Kondo breakdown formalism. In the former one, the critical fluctuations are centered at a small part of the Fermi surface, called hot spots, leaving the majority unaffected and the electrons retaining as Landau quasiparticles. The WF law will be satisfied in this type of QCP due to the integrity of the electrons. In the latter one, the hot spots cover the whole Fermi surface and the critical fluctuations reconstruct the Fermi surface abruptly. The WF law will be violated in this type of QCP due to the breakdown of quasiparticles. The verification of the WF law in $\text{Pr}_2\text{Ir}_2\text{O}_7$ at its QCP unambiguously excludes the possibility of the breakdown of Landau quasiparticles, and is incompatible with the Kondo breakdown formalism. However, this does not immediately indicate a Hertz-Millis type quantum criticality in $\text{Pr}_2\text{Ir}_2\text{O}_7$, because the scaling exponents in the magnetic Grüneisen ratio experiment differ from the expectations within the Hertz-Millis theory. Therefore, new kinds of quantum criticality where the quasiparticle picture is applicable may be realized in $\text{Pr}_2\text{Ir}_2\text{O}_7$, and the confirmation of the WF law puts strong constraints on the description of such QCP.

To better elucidate the significance of the verification of the WF law and its implications to the QCP, we rewrite the third paragraph in Page 5 to:

“Since the thermal conductivity data at low fields collapse on the high-field data below T_s (see Fig. 2b) and the MR is less than 2% for $\mu_0 H \leq 5$ T (see Fig. 1b), it would be inferred that the WF law is obeyed at all the applied fields, even at zero field. This result is of great help to characterize the quantum criticality in $\text{Pr}_2\text{Ir}_2\text{O}_7$. Many efforts have been made to describe the QCP phenomena³⁰, among which two formalisms are highlighted: The Hertz-Millis formalism^{31,32} and the Kondo breakdown formalism³³. In the former one, the critical fluctuations are centered at a small part of the Fermi surface, called hot spots, leaving the majority unaffected and the electrons retaining as Landau quasiparticles. The WF law will be satisfied in this type of QCP due to the integrity of the electrons. In the latter one, the hot spots cover the whole Fermi surface and the critical fluctuations reconstruct the Fermi surface abruptly. The WF law will be violated in this type of QCP due to the breakdown of quasiparticles. The verification of the WF law in $\text{Pr}_2\text{Ir}_2\text{O}_7$ at its QCP unambiguously excludes the possibility of the breakdown of Landau quasiparticles, and is incompatible with the Kondo breakdown formalism. However, this does not immediately indicate a Hertz-Millis type quantum criticality in $\text{Pr}_2\text{Ir}_2\text{O}_7$, because the scaling exponents in the magnetic Grüneisen ratio experiment differ from the expectations within the Hertz-Millis theory¹⁰. Therefore, new kinds of quantum criticality where the quasiparticle picture is applicable may be realized in $\text{Pr}_2\text{Ir}_2\text{O}_7$, and the confirmation of the WF law puts strong constraints on the description of such QCP.”

In the abstract and the summary part, we add a sentence:

“This result puts strong constraints on the description of the quantum criticality in $\text{Pr}_2\text{Ir}_2\text{O}_7$.”

In regards to (2): while the observation isotropic MTC is an interesting result (in view of the anisotropic low-temperature specific heat, for example), the authors really do not provide any real hints as to why it might be so and what is the significance/importance of this result (except, perhaps the statement that the physics of Pr₂Ir₂O₇ deviates from that of spin ice).

While those results are interesting, I find that the authors' inability to provide a springboard as to what and how to think about their two results makes the case for Nature Communications weak. In short, their story for results (1) and (2) just does not go far enough.

We thank Referee #1's valuable suggestion. In our revised manuscript, we have rewritten the discussion part to discuss the possible origin and the significance of the giant isotropic MTC more clearly. We have also proposed experiments that may guide the future studies.

In the first paragraph of the discussion part, we propose that the isotropic MTC in Pr₂Ir₂O₇ indicates that it is not fluctuations from Ising dipole moments but fluctuations from transverse part, likely quadrupole moments, that scatter phonons strongly above T_s . The quantum fluctuations from multipolar interactions are important ingredients for Pr₂Ir₂O₇ to be a QSI candidate. For other Pr-based pyrochlore compounds, both Pr₂Sn₂O₇ and Pr₂Zr₂O₇ show the quantum dynamics and strong quantum fluctuations. However, to our knowledge, the role of quadrupole moments and quantum fluctuations in Pr₂Ir₂O₇ has not been explored experimentally. Our results present the first evidence of quantum effect in this spin liquid candidate and put its Pr 4*f* moments in line with those in other Pr-based QSI candidates.

In the second paragraph of the discussion part, we point out that although Pr₂Ir₂O₇ might lie in the QSI scenario, no contributions from well-defined magnetic excitations are detected in our experiments. This may be a deviation from the QSI scenario despite the presence of quantum fluctuations. We propose that thermal Hall measurements may be helpful to elucidate the true ground state.

In the third paragraph of the discussion part, we discuss the possible role of conduction *d* electrons in the isotropic MTC. Due to the unique metallic nature of Pr₂Ir₂O₇, it is not a good idea to neglect the complex *f-d* coupling between Pr moments and Ir electrons. Therefore, we propose Kondo coupling as another possible origin of isotropic MTC and suggest a potential experiment that may help to check this hypothesis.

The theoretical description of the transport properties of this material is quite challenge. To our knowledge, there are no theoretical works regarding the heat transport phenomena in Pr₂Ir₂O₇, which involve at least three degrees of freedom: phonons, localized *f* moments and conduction *d* electrons. In this context, the exact origin of the isotropic MTC may still be unclear. However, as the first experimental study on the magnetic excitations in Pr₂Ir₂O₇, our results indicate the importance of the quadrupolar interactions and quantum fluctuations. The implication of quantum effect that deviates from the dipole spin-ice physics will shed light on

the future theoretical and experimental studies on determining the true ground state and magnetic excitations in $\text{Pr}_2\text{Ir}_2\text{O}_7$. We hope Referee #1 is satisfied.

Therefore, we have rewritten the discussion part.

Besides, we rewrite the last sentence in abstract to:

“Unexpectedly, although the specific heats are anisotropic with respect to magnetic field directions, the thermal conductivities display the giant but isotropic response. This indicates that quadrupolar interactions and quantum fluctuations are important, which will help determine the true ground state of this material.”

In the summary part, we rewrite the last two sentences to:

“A giant isotropic magneto-thermal conductivity is found at finite temperatures, indicating that the quadrupolar interactions and quantum fluctuations may play important roles. Although further experimental work is required to determine the true ground state and magnetic excitations in $\text{Pr}_2\text{Ir}_2\text{O}_7$, our results suggest the importance of the quantum effect that deviates from the dipole spin-ice physics, and shed light on the future theoretical and experimental studies on this metallic quantum spin liquid candidate.”

I have some specific comments that the authors might want to consider:

- The authors devote quite a bit of space to talk about organized structures in various field directions for spin ice. This is particularly important for the meta magnetic transition in [111] field. I think that the key point here is that at $T \sim 2\text{K}$, there are 2-in/2-out \gg magnetic \ll correlations in place that provide a “holding” field that has to be overcome for the transition. I am not sure how far one can assign spin ice physics in the broadest sense of all its phenomenology to $\text{Pr}_2\text{Ir}_2\text{O}_7$. My point is that, perhaps, the label of (quantum) spin ice for $\text{Pr}_2\text{Ir}_2\text{O}_7$ may be used with some caution.

We agree with Referee #1 that the label of (quantum) spin ice for $\text{Pr}_2\text{Ir}_2\text{O}_7$ may be used with some caution. Indeed, there are no solid experimental evidences that prove $\text{Pr}_2\text{Ir}_2\text{O}_7$ as a quantum spin ice up to now. However, there are some data suggestive of spin-ice correlations between Pr Ising moments. For example, (i) no magnetic order is observed down to the lowest temperature measured, a basic ingredient for a spin ice. (ii) A metamagnetic transition is observed only in [111] field direction, an evidence that Pr Ising moments are fluctuating within the ice manifold. (iii) The spin entropy approaches the Pauling residual entropy at 0.4 K, indicating that the system may enter the spin ice regime. In this context, early experiments have been well explained in the physical picture of classical spin ice + conducting d electrons, such as Ref. 6-8 in our manuscript. It is reasonable to imagine that spin ice physics may play a vital role in $\text{Pr}_2\text{Ir}_2\text{O}_7$.

But it is for sure that (quantum) spin ice is not the whole story, and not all of its phenomena can be assigned to spin ice. For example, (i) the Curie-Weiss temperature is antiferromagnetic, unlike classical spin ice systems. (ii) A quadratic band touching at the Γ point for the Ir 5d electrons is identified, from which various topological phases can emerge. These results imply further correlations besides the spin-ice correlations.

In our manuscript, the label of quantum spin ice is not used as an endorsement that we agree with this description, but as a possible and most intuitive scenario to analyze our data. In fact, as one of our conclusions, this scenario is incomplete and need further study.

- The authors employ the word “spin” in numerous places in their manuscript: in terms of the spin correlations, spin fluctuations, etc.

I find that their usage of the word spin is possibly confusing  what do they mean by “spin”? Do they mean the magnetic (dipole) moment, which should be strictly having a local [111] component for Pr³⁺ being non-Kramers? Or do they refer to the S=1/2 pseudospin? The former would be fine while the latter would be confusing since the transverse parts of the “spin” represent the quadrupole of Pr³⁺.

In this context, it seems that the authors discuss “fluctuations of the spins” and “monopoles” as two separate types of fluctuations (which would be fine in some broad sense if they had in mind the S=1/2 pseudospin description, but in other places, they seem to equate spin to dipole moment. I had commented on that in my previous report, and I still find the authors language unclear.

We feel sorry for the unclear statement of “spin” in our manuscript. Pr³⁺ can be represented by a 1/2-pseudospin. The z component along the local $\langle 111 \rangle$ direction carries a magnetic dipole moment, while the transverse component of the pseudospin corresponds to a quadrupole moment. When discussing the origin of the strong suppression of κ below T_s , we consider two kinds of scattering of phonons, either by well-defined quasiparticles like monopoles or by the fluctuating spins. As for the former one, it refers to the Ising dipole moments pointing along the local $\langle 111 \rangle$ axes. For the latter one, it refers to the transverse fluctuations, a characteristic of quantum spin ice, where quadrupole moments may play a more important role and can induce quantum dynamics.

In our revised manuscript, we have emphasized that the spin refers to the dipole moments when talking about monopoles, and replaced the spin fluctuations with transverse fluctuations in order to avoid confusion. For example, we rewrite the sentence from:

“the strong suppression of κ at low fields apparently comes from the scattering of phonons by the spin system through the spin-lattice coupling, either by well-defined excitations like magnetic monopoles or by fluctuating spins.”

to

“the strong suppression of κ at low fields apparently comes from the scattering of phonons by the spin system through the spin-lattice coupling, either by well-defined excitations like magnetic monopoles, which accompany flips of the magnetic dipole moments pointing along the local $\langle 111 \rangle$ axes, or by transverse fluctuations, possibly including the quadrupole moments of Pr^{3+} .”

In the third paragraph of Page 7, we delete the sentence “Spin freezing has also been observed in classical spin ice like $\text{Dy}_2\text{Ti}_2\text{O}_7$, and the freezing temperature increases with applying field³⁶”, because the freezing in classical spin ice is associated with Ising dipole moments, which is not related to the transverse fluctuations discussed in this paragraph. The freezing in $\text{Pr}_2\text{Ir}_2\text{O}_7$ may be different from that in $\text{Dy}_2\text{Ti}_2\text{O}_7$ since a strong quantum fluctuation is observed.

- In lines [160-162], and [210-212], the authors mention the role of monopoles on the thermal conductivity of $\text{Dy}_2\text{Ti}_2\text{O}_7$ spin ice, but I was unable to grasp what is the point they wish to make about that topic and thermal conductivity (due to monopoles) in $\text{Pr}_2\text{Ir}_2\text{O}_7$.

The purpose to mention the role of monopoles on the thermal conductivity of $\text{Dy}_2\text{Ti}_2\text{O}_7$ and $\text{Ho}_2\text{Ti}_2\text{O}_7$ is to show that the excitations can affect the thermal conductivity in two different ways, either positive contributions or negative contributions, both in classical spin ice and quantum spin ice candidates like $\text{Yb}_2\text{Ti}_2\text{O}_7$ and $\text{Pr}_2\text{Zr}_2\text{O}_7$. This is strongly contrary to the absence of monopole signatures in $\text{Pr}_2\text{Ir}_2\text{O}_7$. Since this paragraph in our revised manuscript focuses on the discussion about the magnetic excitations in QSI candidates, we choose to delete this sentence as $\text{Dy}_2\text{Ti}_2\text{O}_7$ and $\text{Ho}_2\text{Ti}_2\text{O}_7$ are classical spin ices.

- In line #145, the authors write that there is no breakdown of the Landau quasi-particles ... in this non-Fermi liquid. I do not understand how one has fermionic Landau quasi-particles (leading to a satisfied WF Law) but the system being a non-Fermi liquid.

Given nearly a hundred different materials which show non-Fermi liquid behavior due to proximity to a QCP, only several systems display the violation of the WF law without controversies, such as CeCoIn_5 (Ref. 15) and YbAgGe (Ref. 19). In fact, the WF law is quite universal. Even in systems exhibiting obvious hallmarks of non-Fermi liquids, like CeNi_2Ge_2 (Ref. 20) and $\text{Sr}_3\text{Ru}_2\text{O}_7$ (Ref. 21), the WF law is observed to hold. The reason is explained in the response to the first comment. In the Hertz-Millis theory, the quantum critical fluctuations are centered at a small part of the Fermi surface, leaving the majority unaffected and the electrons retaining as Landau quasiparticles. Thus, it is not surprising that the Landau quasiparticles persist in a non-Fermi liquid.

- The authors discuss the anisotropic nature of the specific heat while the MTC is isotropic. I found that the authors do not adequately discuss the link (really, paradox) of having anisotropic C_v but isotropic MTC. In the same vein, the discussion in lines [174-178] could have been clearer. In particular, for $\text{Dy}_2\text{Ti}_2\text{O}_7$, the different field directions allow the Dy moments to develop phase transitions by decoupling some of the Ising moments (as

originally proposed in Nature 399, 333 [1999] and explained in Phys. Rev. Lett. 95, 097202 [2015]).

We appreciate Referee #1's valuable suggestion. In our revised manuscript, we discuss the link between the anisotropic specific heat and isotropic MTC more clearly by interpreting that anisotropic specific heats will normally lead to anisotropic phonon thermal conductivities.

In the second paragraph of Page 7, we add a sentence:

“As a result, the scattering rate between phonons and monopoles Γ_{p-m} should be anisotropic, if one only considers the classical spin ice scenario originating from the local dipole moments. This leads to an anisotropic phonon thermal conductivity $\kappa_p=1/3C_p v_p^2/(\Gamma_{p-m}+\Gamma_{other})$, where C_p , v_p , Γ_{other} , are the phonon specific heat, phonon velocity and scattering rate from other mechanism like defects, respectively. ”

In the same paragraph, we add a sentence to discuss the reason of anisotropic specific heats in $Dy_2Ti_2O_7$:

“Especially, the different field directions allow the Dy moments to develop phase transitions by decoupling some of the Ising moments^{40,41}. ”

- Unless I missed it, I could not find what is the field direction for the thermal conductivity measurement with $H \perp [111]$?
Is it $[1 -1 0]$ or some other direction?

The magnetic field direction labelled $H \perp [111]$ is schematically illustrated in the Fig. 4a and 4b. It is parallel to the largest sample surface, which is (111) plane. Since the thermal conductivities are all isotropic with respect to all the measured magnetic field direction, we did not determine the detailed direction.

- In line [216], the authors write “this might indicate the quantum dynamics of spins in Pr2Ir2O7”. Here, again, the word spin is used and it is not clear at all what is the point/logic the authors aim to make by this statement.

We agree with Referee #1's comment. This sentence is not very relevant to this paragraph. We delete this sentence in the revised manuscript.

Generally speaking, I find the whole discussion section pretty much void of useful information that might give a possible/plausible suggestion as to what may be going on with the isotropic MTC in Pr2Ir2O7 — It's pretty much 1.5 page of text that does not say much (including lines 247-250) and, in my opinion, this is why this paper should not be published in Nature Communications as, to no fault to the authors, they really don't seem to have an idea as to why the MTC is isotropic.

If the MTC came from the electrons (if they were not proper Landau quasiparticles), then one might think the MTC would “have more chance” to be more isotropic than a “spin/lattice” coupling channel. But the authors have ruled this out in the first part of their results section. In that vein, unless I missed it, the authors do not seem to tie together their WF results with their MTC results. I provided one such connection here, but maybe they see others (and better ones). If so, they should mention them.

We have rewritten the discussion part in our manuscript to discuss the possible origin and the significance of the giant isotropic MTC more clearly. Please see the detailed response above. Again, although the exact origin of the isotropic MTC may still be unclear, as the first experimental study on the magnetic excitations in $\text{Pr}_2\text{Ir}_2\text{O}_7$, our results indicate the importance of the quadrupolar interactions and quantum fluctuations. The implication of quantum effect that deviates from the dipole spin-ice physics will shed light on the future theoretical and experimental studies on determining the true ground state and magnetic excitations in $\text{Pr}_2\text{Ir}_2\text{O}_7$.

- Minor comment: what is the meaning of the subscript “s” in T_s ?

The subscript “s” in T_s means “suppression”. It means that the thermal conductivities are strongly suppressed above certain temperatures. We denote these temperatures as suppression temperatures.

Referee #2:

In this paper, the ultralow-temperature thermal conductivity of $\text{Pr}_2\text{Ir}_2\text{O}_7$ single crystals was investigated. The present paper is worth publishing, but I asked high-temperature thermal conductivity data above 1 K. According to my suggestion, the authors have carried out the magneto-thermal conductivity measurements at high temperatures up to 50 K and discussed the data. Therefore, the revised paper can be published.

We appreciate Referee #2's recommendation.

Referee #3:

I have read the re-submitted manuscript as well as the authors' response to both my comments and those of the other two referees. My original review of the manuscript was reasonably favourable, but I required commentary regarding the possible nature of the Ir magnetism in generating the isotropic response that is reported. I felt that the authors provided a reasonable response to my point. The most detailed comments appear to come from Ref. 1 - again at the level that I can tell, their response seems to be reasonable. For these reasons I now recommend acceptance as a Nature Communications.

We appreciate Referee #3's recommendation.

List of changes to manuscript

1) Page 1, the abstract:

We add a sentence to state the significance of the verification of WF law:

“This result puts strong constraints on the description of the quantum criticality in $\text{Pr}_2\text{Ir}_2\text{O}_7$.”

2) Page 1, the abstract:

We rewrite the last sentence to state the significance of the result of isotropic MTC to:

“Unexpectedly, although the specific heats are anisotropic with respect to magnetic field directions, the thermal conductivities display the giant but isotropic response. This indicates that quadrupolar interactions and quantum fluctuations are important, which will help determine the true ground state of this material.”

3) Page 2, the third paragraph:

We rewrite the last sentence to:

“from which knowledge about the role of multipolar interactions beyond the dipolar interactions and quantum fluctuations can be obtained.”

4) Page 3, the second paragraph:

We replace the words “fluctuating spins” to “transverse fluctuations”.

5) Page 5, the third paragraph:

We rewrite the paragraph to elucidate the significance of the verification of the WF law and its implications to the QCP:

“Since the thermal conductivity data at low fields collapse on the high-field data below T_s (see Fig. 2b) and the MR is less than 2% for $\mu_0 H \leq 5$ T (see Fig. 1b), it would be inferred that the WF law is obeyed at all the applied fields, even at zero field. This result is of great help to characterize the quantum criticality in $\text{Pr}_2\text{Ir}_2\text{O}_7$. Many efforts have been made to describe the QCP phenomena³⁰, among which two formalisms are highlighted: The Hertz-Millis formalism^{31,32} and the Kondo breakdown formalism³³. In the former one, the critical fluctuations are centered at a small part of the Fermi surface, called hot spots, leaving the majority unaffected and the electrons retaining as Landau quasiparticles. The WF law will be satisfied in this type of QCP due to the integrity of the electrons. In the latter one, the hot spots cover the whole Fermi surface and the critical fluctuations

reconstruct the Fermi surface abruptly. The WF law will be violated in this type of QCP due to the breakdown of quasiparticles. The verification of the WF law in $\text{Pr}_2\text{Ir}_2\text{O}_7$ at its QCP unambiguously excludes the possibility of the breakdown of Landau quasiparticles, and is incompatible with the Kondo breakdown formalism. However, this does not immediately indicate a Hertz-Millis type quantum criticality in $\text{Pr}_2\text{Ir}_2\text{O}_7$, because the scaling exponents in the magnetic Grüneisen ratio experiment differ from the expectations within the Hertz-Millis theory¹⁰. Therefore, new kinds of quantum criticality where the quasiparticle picture is applicable may be realized in $\text{Pr}_2\text{Ir}_2\text{O}_7$, and the confirmation of the WF law puts strong constraints on the description of such QCP. ”

6) Page 6-8, the Giant isotropic MTC and its origin part:

We rewrite this part to emphasize that the spin refers to the dipole moments when talking about monopoles, and replace the spin fluctuations with transverse fluctuations in order to avoid confusion.

7) Page 8-10, the Discussion part:

We rewrite the discussion part to discuss the possible origin and the significance of the giant isotropic MTC more clearly.

8) Page 10, the Summary paragraph:

We rewrite the summary paragraph to state the significance of the verification of WF law and the isotropic MTC to:

“In summary, we have measured the ultralow-temperature thermal conductivity of $\text{Pr}_2\text{Ir}_2\text{O}_7$ single crystals. The Wiedemann-Franz law is verified at high magnetic fields and inferred at zero field, suggesting the normal behavior of electrons at the zero-field quantum critical point and the absence of mobile fermionic magnetic excitations. This result puts strong constraints on the description of the quantum criticality in $\text{Pr}_2\text{Ir}_2\text{O}_7$. A giant isotropic magneto-thermal conductivity is found at finite temperatures, indicating that the quadrupolar interactions and quantum fluctuations may play important roles. Although further experimental work is required to determine the true ground state and magnetic excitations in $\text{Pr}_2\text{Ir}_2\text{O}_7$, our results suggest the importance of the quantum effect that deviates from the dipole spin-ice physics, and shed light on the future theoretical and experimental studies on this metallic quantum spin liquid candidate.”

9) Several references are added:

30. v. Löhneysen, H., Rosch, A., Vojta, M. & Wölfle, P. Fermi-liquid instabilities at magnetic quantum phase transitions, *Rev. Mod. Phys.* **79**, 1015 (2007).

31. Hertz, J. A. Quantum critical phenomena. *Phys. Rev. B* **14**, 1165 (1976).

32. Millis, A. J. Effect of a nonzero temperature on quantum critical points in itinerant fermion systems. *Phys. Rev. B* **48**, 7183 (1993).
33. Si, Q., Rabello, S., Ingersent, K. & Smith, J. L. Locally critical quantum phase transitions in strongly correlated metals. *Nature* **413**, 804 (2001).
40. Ramirez, A. P., Hayashi, A., Cava, R. J., Siddharthan, R. & Shastry, B. S. Zero-point entropy in 'spin ice', *Nature* **399**, 333 (1999).
41. Ruff, J. P. C., Melko, R. G. & Gingras, M. J. P. Finite-Temperature Transitions in Dipolar Spin Ice in a Large Magnetic Field, *Phys. Rev. Lett.* **95**, 097205 (2005).
46. Onoda, S. & Tanaka, Y. Quantum Melting of Spin Ice: Emergent Cooperative Quadrupole and Chirality, *Phys. Rev. Lett.* **105**, 047201 (2010).
47. Zhou, H. D., Wiebe, C. R., Janik, J. A., Balicas, L., Yo, Y. J., Qiu, Y., Copley, J. R. D. & Gardner, J. S. Dynamic Spin Ice: $\text{Pr}_2\text{Sn}_2\text{O}_7$, *Phys. Rev. Lett.* **101**, 227204 (2008).
48. Kimura, K., Nakatsuji, S., Wen, J.-J., Broholm, C., Stone, M. B., Nishibori, E. & Sawa, H. Quantum fluctuations in spin-ice-like $\text{Pr}_2\text{Zr}_2\text{O}_7$, *Nat. Commun.* **4**, 1934 (2013).
49. Hirschberger, M., Krizan, J. W., Cava, R. J. & Ong, N. P. Large thermal Hall conductivity of neutral spin excitations in a frustrated quantum magnet, *Science*, **348**, 106 (2015).